Corrected: Author correction

# Dynamical nonlinear memory capacitance in biomimetic membranes

Joseph S. Najem [1,2], Md Sakib Hasan[3], R. Stanley Williams [4], Ryan J. Weiss[3], Garrett S. Rose[3], Graham J. Taylor [1,5], Stephen A. Sarles[1] & C. Patrick Collier[6]

Two-terminal memory elements, or memelements, capable of co-locating signal processing and memory via history-dependent reconfigurability at the nanoscale are vital for next-generation computing materials striving to match the brain's efficiency and flexible cognitive capabilities. While memory resistors, or memristors, have been widely reported, other types of memelements remain underexplored or undiscovered. Here we report the first example of a volatile, voltage-controlled memcapacitor in which capacitive memory arises from reversible and hysteretic geometrical changes in a lipid bilayer that mimics the composition and structure of biomembranes. We demonstrate that the nonlinear dynamics and memory are governed by two implicitly-coupled, voltage-dependent state variables—membrane radius and thickness. Further, our system is capable of tuneable signal processing and learning via synapse-like, short-term capacitive plasticity. These findings will accelerate the development of low-energy, biomolecular neuromorphic memelements, which, in turn, could also serve as models to study capacitive memory and signal processing in neuronal membranes.

[1] Department of Mechanical, Aerospace and Biomedical Engineering, University of Tennessee, Knoxville, TN 37916, USA. [2] Joint Institute for Biological Sciences, Oak Ridge National Laboratory, Oak Ridge, TN 37831, USA. [3] Department of Electrical Engineering and Computer Science, University of Tennessee, Knoxville, TN 37916, USA. [4] Department of Electrical and Computer Engineering, Texas A&M University, College Station, TX 77840, USA. [5] Bredesen Center for Interdisciplinary Research, University of Tennessee, Knoxville, TN 37996, USA. [6] Center for Nanophase Materials Sciences, Oak Ridge National Laboratory, Oak Ridge, TN 37831, USA. Correspondence and requests for materials should be addressed to S.A.S. (email: ssarles@utk.edu) or to C.P.C. (email: colliercp@ornl.gov)

Neuromorphic computing systems striving to match the density, interconnectivity, and efficiency of the brain require highly parallelized networks of dynamic elements and materials that co-locate signal processing and memory—just like biological synapses[1–4]. Although three-terminal transistors are being used[5–8] for this purpose, they require complex circuits to emulate synapses, are power hungry[9], and their scaling is limited by the imminent end of Moore's Law[3,10]. Two-terminal memory elements, i.e., memelements[1,11–14], that are able to store and process information through history-dependent material reconfigurations at the nanoscale, offer an alternative path to greater functional density and reduced energy consumption. By definition, memelements are devices with resistance, capacitance, or inductance that depend on their past electrical activity[14]. While many types of memory resistors, i.e., memristors, have been introduced[15–18], to date, only a sparse collection of physically realized memory capacitors[19–30] and one pseudo-memcapacitor[31] have been reported, despite their promise to further lower static power dissipation. Of these, most did not provide evidence of pinched hysteresis in the charge-voltage plane as proof of memory capacitance, as first defined by Chua[14,32], or develop realistic, physics-based models to describe and predict the state variables driving capacitive reconfigurations. Devices for which ideal, analogue memcapacitance that originates from geometrical changes in materials at the molecular scale remain unrealized to date.

We recently demonstrated that a synthetic biomembrane (i.e., lipid bilayer) doped with voltage-activated ion channels exhibits volatile memory resistance governed by two voltage-dependent state variables: the areal density of ion channels and the increase in membrane area due to electrowetting[16]. We have also shown that the spike-rate-dependent plasticity (SRDP) exhibited by these two-terminal memristors enable them to function as resistive synapses for online learning in spike recurrent neural networks built from solid-state neurons[33,34]. Based on these findings we hypothesized that an insulating lipid bilayer, without conductive ion channels, may exhibit capacitive memory governed solely by voltage-dependent changes to the dimensions of the dielectric dominated by the hydrophobic core of the bilayer.

Here we report that adhering, lipid-encased, aqueous droplets in oil yields an interfacial biomimetic membrane (3–5 nm thick) that exhibits volatile, analogue memcapacitance via voltage-controlled geometric reconfigurability. Pinched hysteresis in both the charge-voltage and capacitance-voltage planes result from dynamic changes in interfacial area and hydrophobic thickness, each of which are nonlinearly dependent on voltage. Through experimentation and modelling, we demonstrate this assembly is a volatile, second-order, generic memcapacitor[32] capable of synapse-like temporal filtering and learning through short-term plasticity. Our results forecast new classes of biomimetic, low-power memelements based on soft, organic materials and biomolecules, which, in turn, will aid in exploring capacitive memory and susceptibility in neuronal membranes.

## Results

**A two-terminal biomolecular memcapacitor.** Inspired by plasma membranes (Supplementary Note 1)[35], we have developed a two-terminal, biomimetic assembly with dynamical, nonlinear memory capacitance via voltage-controlled geometric reconfigurability. The system consists of an elliptical, planar lipid bilayer that forms at an interface between two lipid-coated aqueous droplets (~200 nL each) in oil (Fig. 1a and Supplementary Figs. 1–3). With this structure, the hydrophobic lipid acyl chains and any residual oil (each with a dielectric constant of ~2–3) form a low-leakage (membrane resistance > 100 MΩ cm²)[16,36], parallel-plate capacitor (0.1–1 μF cm⁻²)[37] that inhibits diffusive transport between droplets. As is well established, the series capacitance formed by the electric double layer (~75 μF cm⁻²)[38] of ions on each face of the membrane can be ignored since it is much larger in value than that of the hydrophobic region of the membrane (~0.4–0.65 μF cm⁻²; Supplementary Note 2). The assembly process, interface geometry, and physical properties are detailed in Supplementary Note 3 and elsewhere[16,37]. We use synthetic 1,2-diphytanoyl-*sn*-glycero-3-phosphocholine (DPhPC) lipids, known for their chemical stability and low ion permeability[39], to construct capacitive bilayer interfaces, though other lipid types[40,41] or polymers[42] may also be used. Membranes are assembled in decane ($C_{10}H_{22}$) and hexadecane ($C_{16}H_{34}$) oils to study the effect of alkane length and viscosity (0.86 and 3.04 mN s m⁻², respectively)[43] on voltage-dependent bilayer geometry. At zero membrane potential, the specific capacitance of a DPhPC bilayer in decane is 0.46 μF cm⁻² (equivalent hydrophobic thickness, $W_0 \sim 3.8$ nm), versus 0.68 μF cm⁻² ($W_0 \sim 2.6$ nm) in hexadecane (Supplementary Fig. 4). A thicker bilayer with decane results from more residual oil in the membrane (~43% by volume) due to the shorter chain length of decane[44] (versus ~10% for hexadecane[45]). By comparison, the specific capacitance of a neural membrane[46] is ~0.90 μF cm⁻². Further, the zero-volt minor axis radii ($R_0$) of elliptical interfaces formed in decane and hexadecane are ~100 μm and ~250 μm, respectively (see Supplementary Movies 1–3), due to oil-dependent differences in monolayer tensions[37] (see Supplementary Notes 3–5 and Supplementary Fig. 3).

Subjected to a nonzero bias, $v(t)$, ions rapidly accumulate on both sides of the membrane where they exert parallel and perpendicular forces that can affect the geometry of the interface. For a typical bilayer with nominal capacitance of ~500–1000 pF and aqueous droplets with solution resistance of ~10 kΩ, the time constant for charging or discharging the membrane is ~10 μs. Therefore, at frequencies well below ~15 kHz, ion movements within the electric double layer are quasi-static with respect to AC voltage. Voltage-driven geometrical changes can include: (1) the formation of new bilayer area at constant thickness[37] due to charge-induced reductions in bilayer tension—electrowetting (EW)[37]—described by the Young-Lippmann equation[47] (Supplementary Movies 4 and 5); and (2) a decrease in thickness—electrocompression (EC)[16]—that may or may not affect bilayer area. In oil-free membranes, EC of the hydrophobic acyl chains is explicitly coupled to increased area at constant volume, due to the Poisson effect[48]. However, EC of bilayers containing residual oil trapped between lipid leaflets (e.g., DPhPC bilayers in decane or hexadecane) expels the oil rather than deforming the acyl chains. Thus, changes in thickness do not affect membrane area (i.e., no stretching)[45] (see Supplementary Note 5 and Supplementary Fig. 4).

Because radius, $R(t)$, and hydrophobic thickness, $W(t)$ are affected via EW and EC in our system, we expect the relationship between charge, $Q$, and applied voltage, $v$, to be described by

$$Q = C(R, W)v \tag{1}$$

where $R$ and $W$ are voltage-controlled state variables, and the memcapacitance, $C$, for a parallel-plate capacitor with planar ellipticity, $a$, and equivalent dielectric constant, $\varepsilon$, is given by

$$C(R, W) = \frac{\varepsilon\varepsilon_0\left(a\pi R(t)^2\right)}{W(t)}. \tag{2}$$

Because the dielectric constants of decane, hexadecane, and the hydrophobic core of the bilayer (2, 2.09, and 2.1 respectively) are very similar, we do not anticipate significant changes in the effective dielectric constant upon EC-induced thinning that

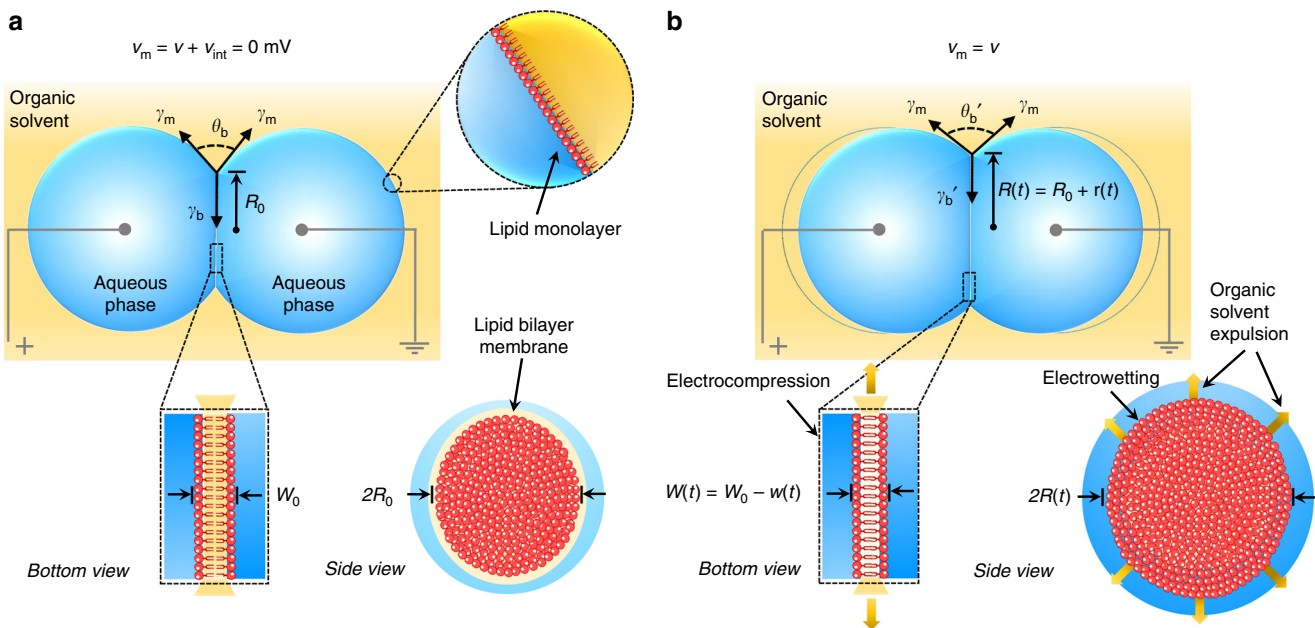

**Fig. 1** Biomimetic membrane assembly and electromechanical behaviours. **a** A capacitive planar lipid bilayer that mimics the structure of a biological membrane forms spontaneously upon contact between lipid-coated droplets and exclusion of excess oil. The elliptical interface represents an equilibrium in adhesive forces governed by: (1) a balance of monolayer, $\gamma_m$, and bilayer, $\gamma_b$, tensions prescribed by Young's equation (Supplementary Note 2, Eq. S.2); and (2) the slight sagging of droplets caused by the water-oil density difference. In the absence of a net membrane voltage, the geometry of the bilayer is described by the zero-volt minor axis radius, $R_0$ (~100–300 μm, determined by analysis of bottom-view, bright-field images (Supplementary Fig. 2), and the hydrophobic thickness, $W_0$ (~2–4 nm). Wire-type (~125 μm diameter) silver/silver chloride (Ag/AgCl) electrodes inserted into the droplets were used to apply a transmembrane voltage and measure the induced ion current. Aqueous droplets (pH 7) contained 500 mM potassium chloride and 10 mM MOPS (3-(N-morpholino)propanesulfonic acid). We define the membrane voltage, $v_m$, as the summation of the applied voltage, $v(t)$, and the intrinsic membrane potential, $v_{int}$ (equal to zero for symmetrical membranes). **b** A schematic describing the geometrical changes caused by a net membrane voltage, $v(t)$. Changes are manifested by EW-driven creation of new bilayer area between opposing lipid monolayers (at constant thickness) and an independent decrease in hydrophobic thickness due to EC-driven removal of residual oil in the membrane. Since the volumes of both droplets remain constant, the external contact angle, $\theta_b'$, and bilayer radius, $R(t)$, increase as EW reduces bilayer tension, $\gamma_b'$. Monolayer tension, $\gamma_m$, is independent of transmembrane voltage (Supplementary Fig. 3)

depletes the amount of oil between lipid leaflets. Based on the full derivation in Supplementary Notes 6 and 7, the dynamical state equation for $R$ is given by

$$\frac{dR(t)}{dt} = \frac{1}{\xi_{ew}}\left(\frac{a\varepsilon\varepsilon_0}{2W(t)}v(t)^2 - k_{ew}(R(t)-R_0)\right) \quad (3)$$

where $\xi_{ew}$ and $k_{ew}$ are the effective damping (N s m$^{-2}$) and stiffness (N m$^{-2}$) coefficients, respectively, in the tangential direction. Similarly, the state equation for $W$ is given by

$$\frac{dW(t)}{dt} = \frac{1}{\xi_{ec}}\left(-\frac{\varepsilon\varepsilon_0 a\pi R(t)^2}{2W(t)^2}v(t)^2 + k_{ec}(W_0-W(t))\right) \quad (4)$$

where $\xi_{ec}$, and $k_{ec}$ are the effective damping (N s m$^{-1}$) and stiffness (N m$^{-1}$) coefficients, respectively, in the normal direction. These nonlinear, implicitly-coupled state equations stem from electrical forces exerted on the membrane that are themselves functions of $R$ and $W$ (see Supplementary Note 7 and Supplementary Fig. 5).

**Dynamical response and pinched hysteresis.** We first measure the dynamical capacitance of DPhPC lipid membranes in response to a bipolar alternating bias, $v(t)$, at room temperature (RT ~ 22 °C) (Fig. 2). These data reveal basic information about the dynamical responses of our biomimetic system: First, bilayer capacitance exhibits a positive DC offset at steady state, following an exponential transient period of 2–20 s, induced by the root mean square of the voltage. Second, dynamical capacitance shows a doubling of the excitation frequency due to the squaring of the

membrane voltage in Eqs. 3, 4, indicating the process is proportional to the power and independent of the sign of the applied AC voltage. Third, for symmetric membranes at steady state, the capacitance is minimum at zero applied potential (Fig. 3a, b) and the amplitude of capacitance oscillations decreases with increasing excitation frequency, consistent with Chua's explanation that memristor pinched hysteresis collapses to a simple curve (Fig. 3) at high frequency because the system cannot follow the driving excitation[14]. Finally, these measurements highlight the fact that the capacitance of a bilayer formed in decane is more responsive to voltage compared to one in hexadecane, especially at 1.7 Hz (Fig. 2d), due to the reduced viscosity of decane and the fact that greater increases in area and decreases in thickness are possible due to higher zero-volt thickness and zero-voltage monolayer tension of DPhPC membranes in decane that make them more susceptible to EC and EW.

To investigate the memcapacitance, we plot both the steady-state dynamic capacitance versus voltage (C-v) and charge versus voltage (Q-v) relationships (Fig. 3, see "Methods"). For both oils, we observe: (1) symmetric, nonlinear, nonhysteretic relationships at lower frequencies (<0.005 Hz) (Supplementary Fig. 6); (2) symmetric, nonlinear, pinched hysteresis Q-v loops at intermediate frequencies (Fig. 3c, d); and (3) symmetric, linear, nonhysteretic relationships implying fixed capacitance, $C$, at higher frequencies (>5 Hz and >2 Hz for decane and hexadecane, respectively) (Figs. 2, 3).

The nonlinear changes and pinched hysteresis in Q-v responses at low and intermediate frequencies arise from reversible, hysteretic changes in $C$ (Fig. 3a, b and upper insets of Fig. 3c, d), caused by

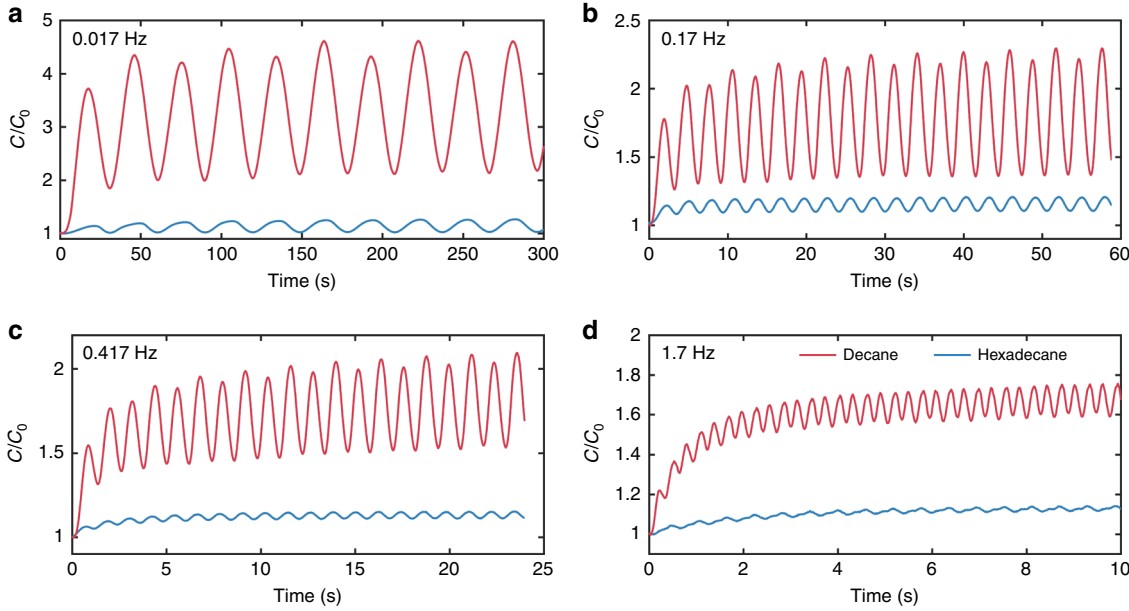

**Fig. 2** Dynamical changes in membrane capacitance in response to sinusoidal voltage excitation. Governed by EW and EC, the capacitances of symmetrical DPhPC membranes formed either in decane or hexadecane increase in response to bipolar sinusoidal voltages ($|150\,mV|$). Based on our experimental measurements, we found that for both oils and at all frequencies (**a–d**), DC capacitance exhibits a damped increase to a positive steady-state offset governed by the RMS value of the signal

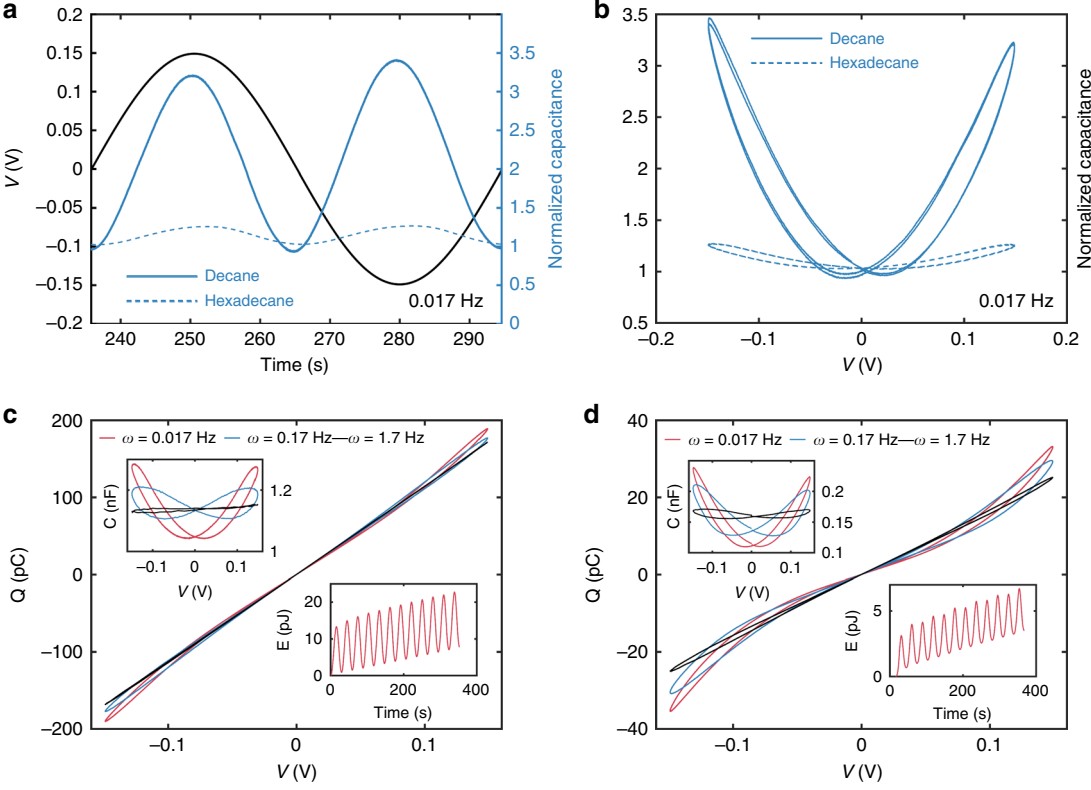

**Fig. 3** Frequency dependence of pinched hysteresis for DPhPC memcapacitors. **a**, **b** Display the dynamical capacitance of a symmetric DPhPC bilayer as a function of time and voltage, respectively. The results show that steady-state dynamic capacitance is minimum when $v$ is zero. As expected, EW and EC incur significantly greater changes in normalized capacitance ($C/C_0$) for a DPhPC bilayer in decane. **c**, **d** Display $Q$-$v$ and $C$-$v$ (upper insets) measured in response to a sinusoidal voltage, $v(t)$, applied to DPhPC bilayer memcapacitors assembled in hexadecane and decane, respectively. The lower insets in each display the energy dissipated by the reconfigurable membrane at an excitation frequency of 0.017 Hz. Pinched hysteresis for bilayers in hexadecane occurs at excitation frequencies between 0.005–2 Hz, whereas bilayers in decane exhibit pinched hysteresis across a wider frequency range, from 0.001–5 Hz. These results highlight the system's modularity, where changing the oil can be used to tune the bandwidth of memory capacitance

EW and EC as described in Fig. 1b and Eqs. 2–4. These bipolar sweeps also show that a DPhPC bilayer in decane displays larger relative changes in $C$-$v$, and both stronger nonlinearity and more pronounced hysteresis in $Q$-$v$ compared to hexadecane. Membranes in decane also exhibit a wider range of intermediate frequencies where pinched hysteresis in $Q$-$v$ occurs. The former makes sense because oil-rich membranes display higher tensions and thicknesses, conditions where EW and EC can create larger geometrical changes (Supplementary Fig. 7)[44] compared to oil-poor bilayers with weaker voltage dependencies (Supplementary Fig. 8). The latter is consistent with the fact that lower oil viscosity allows changes to occur more quickly (Supplementary Fig. 4). We also obtain $Q$-$v$ measurements for a bilayer in hexadecane at 50 °C (Supplementary Fig. 9) to demonstrate that oil viscosity impacts hysteresis separately from alkane tail length. As expected, the heated system exhibits more pronounced $Q$-$v$ hysteresis compared to room temperature. The lower insets in Fig. 3c, d display the memory-associated energy (Methods) dissipated in response to a sinusoidal $v(t)$ at 0.017 Hz. These energy results demonstrate that a lipid bilayer memcapacitor is a passive device that dissipates energy to drive geometrical changes at the interface and dynamically reconfigure $C$. And because changes in geometry and $C$ are not stored (i.e., $R$ and $W$ return to $R_0$ and $W_0$, respectively, when the voltage is removed), this system exhibits volatile memcapacitance.

We also measure $C$ induced by stepwise changes in voltage from 0 to 150 mV (0.025 Hz, 50% duty cycle) to estimate time constants for increases and decreases in $C$ (Supplementary Figure 7). For both oils, increases in $C$ are slower ($\tau_{rise} \sim 2.2$ s and 3.7 s for decane and hexadecane, respectively) than decreases in $C$ ($\tau_{decay} \sim 0.46$ s and 1.59 s, respectively). Faster decreases in $C$ are helpful in creating pinched hysteresis during decreasing voltage portions of a sweep. Not only are $\tau_{rise}$ and $\tau_{decay}$ for decane smaller than for hexadecane, but the ratio $\tau_{rise}/\tau_{decay}$ is higher. Furthermore, we found that the percentage increase in $C$ in hexadecane matches that of the membrane area (~25%), indicating that only EW causes the change in $C$, and EC is negligible. Conversely, a bilayer in decane exhibited an increase in $C$ of 150%, even though the bilayer area only increased by 80%. This discrepancy suggests $W$ had to have decreased by ~28% to create the additional 70% rise in net capacitance. Comparing the dynamic changes in $C$ to membrane area (Supplementary Fig. 7) also reveals that EW ($\tau_1 \sim 2.2$ s) results in faster increases in $C$ than EC ($\tau_2 \sim 16$ s). Coincidentally these time constants for geometric reconfiguration are quite similar to the time constants (~1 s) attributed to hysteretic ion rearrangement on the surfaces of charged, glass nanopores[49,50], despite the differences in physical origins for hysteretic charging versus ac voltage. Note that if ion channels were present in the lipid membrane, the $Q$-$v$ relationship would resemble that of a leaky capacitor (i.e., hysteretic, but not pinched due to additional ohmic current). Still, the capacitive memory will be present as both EW and EC phenomena remain.

Unlike many memelements, our system is highly modular and uniquely bio-inspired. By changing the composition of one leaflet of the bilayer from DPhPC to DOPhPC (1,2-di-O-phytanoyl-sn-glycero-phosphocholine) lipids (Fig. 4), we create an asymmetric bilayer with a constant intrinsic bias, analogous to the resting potential of a cell (~−70 mV)[51], produced by the difference of lipid dipole potentials[52]. This means when $v = 0$, the net membrane potential, $v_m = v + v_{int}$, equals $v_{int}$ (~−137 mV), which results in a thinner membrane ($W \sim 3.5$ nm) with larger interfacial area ($R \sim 170$ μm), due to EC and EW, compared to symmetric leaflets for which $v_m = v_{int} = 0$. The value of $v$ for which the effects of EW and EC on $C$ are minimized is equal to

+ 137 mV, not zero (Fig. 4a, b and Supplementary Fig. 10). Plotting the $Q$-$v$ relationships for the asymmetric devices (Fig. 4c, d) displays asymmetric pinched-hysteresis loops, where the largest hysteresis lobes occur in the third quadrant where the magnitudes of $v$ and $v_{int}$ add.

**Signal processing via short-term plasticity.** Storing and processing information in the brain depends on continuously changing the strength of synaptic communication between neurons, a process known as synaptic plasticity. Ubiquitous in the brain, short-term synaptic plasticity (STP) is thought to play a significant role in information transfer and neural processing, including spatiotemporal filtering that enables selective transmission of specific patterns of neural activity[53]. Presynaptic STP can last from milliseconds to minutes and is often divided into three categories: depression, facilitation, and augmentation. Facilitating and augmenting synapses act as high-pass filters by increasing their conductances during subsequent bursts of incoming signals, which strengthen communication between connected neurons[53]. On the other hand, depressing synapses function as low-pass filters by reducing their conductances during bursts of activity, resulting in lower transmission rates and connection strengths between neurons.

To investigate short-term plasticity in purely capacitive, lipid-only interfaces, we stimulate them with trains of voltage pulses and record their dynamic current responses (Fig. 5). We discover that for a fixed ON-time (2 ms), and a range of OFF-times (1–50 ms), bilayers accumulate changes in capacitance, causing sequential increases in peak capacitive current ($I_{peak} = C \cdot dv/dt$), which emulate high-pass filtering via short-term facilitation in synapses[53]. Figure 5b–e shows responses of DPhPC bilayers to 150 mV, 2 ms ON pulses separated by 0 mV, 1 ms OFF periods. For all cases, peak capacitive currents increased monotonically across successive pulses until reaching steady state (Supplementary Fig. 11 provides total current responses). Consistent with Supplementary Figure 7, bilayers in decane exhibited an initial faster adaptation to the stimulus ($\tau_1 \sim 2$ s) caused by EW, followed by a slower adaptation ($\tau_2 \sim 16$ s) linked to EC. In contrast, bilayers in hexadecane exhibited only one time constant ($\tau \sim 3.7$ s), attributed to EW.

As discussed earlier, asymmetric devices can exhibit minimum capacitance at a nonzero applied potential. The consequence of this shift is that an asymmetric DPhPC:DOPhPC membrane exhibits a depression-like cumulative decrease in peak current (Fig. 5e) in response to a pulse stimulation of $v = + 150$ mV ($v_m = v + v_{int} = + 13$ mV)—emulating another vital type of STP[54]. Moreover, short-term learning responses also become asymmetric with respect to polarity: applying −150 mV causes facilitatory behaviour, as the magnitudes of both $v$ and $v_{int}$ sum ($v_m \sim −287$ mV) to increase $C$ (Fig. 5e). Facilitating responses are also possible for positive $v$ by sufficiently raising the pulse amplitude; however, dielectric breakdown is common for $|v_m| > 300$ mV.

Figure 5f compares facilitation-like responses versus stimulus interval from symmetric membranes in decane and hexadecane. These signal processing features suggest this type of device could be tuned for use as a high-pass or low-pass filter, or as a memcapacitive synapse with online short-term learning capabilities[31,54]. We are currently combining multiple symmetric and asymmetric leaflet membranes to achieve facilitation followed by depression, and vice versa.

**Simulations of memory capacitance.** After confirming volatile memcapacitance, we applied a simultaneous, nonlinear least squares fitting routine (Supplementary Methods) to independent, representative measurements of both $C$ and $R$ induced by sine

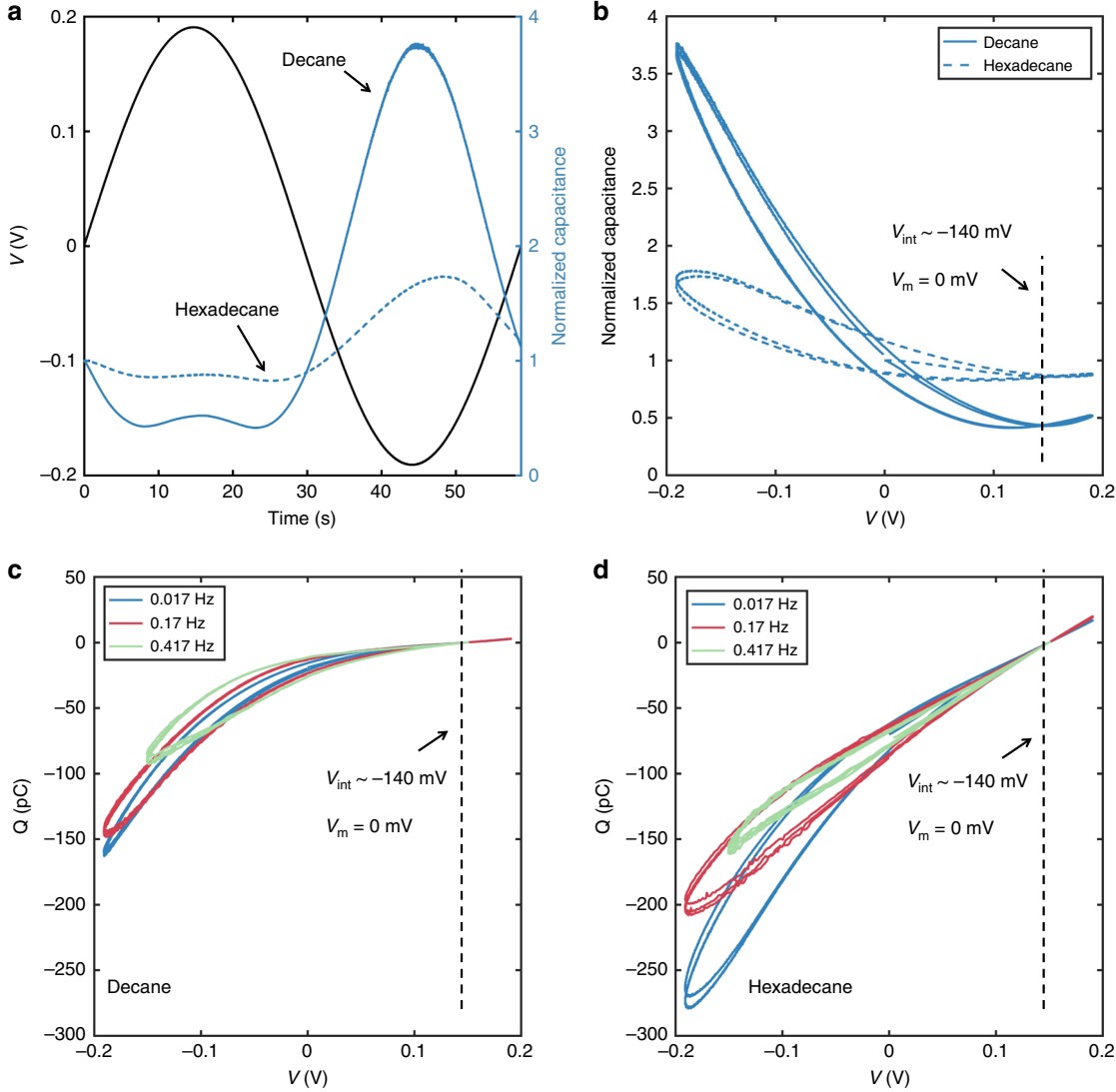

**Fig. 4** Dynamical capacitance as a function of time and voltage for an asymmetric DPhPC:DOPhPC membrane. **a, b** show that, for both oils, minimum capacitance occurs at −137 mV, which is equal in magnitude and opposite in sign to the intrinsic membrane potential created by the difference in dipole potentials. Unlike in symmetric membranes, where $v_m = v$, the membrane voltage for a symmetric bilayer is given by, $v_m = v + v_{int}$. This explains the asymmetry in the peak capacitance values at +/− 150 mV. We also plot the $Q$-$v$ relationships for asymmetric devices formed in either decane (**c**) and hexadecane (**d**). The results show asymmetric pinched-hysteresis loops, where the largest hysteresis lobes occur in the third quadrant where the magnitudes of $v$ and $v_{int}$ sum. Note that the hysteresis for a membrane in hexadecane (**d**), appears larger than that of a bilayer in decane (**c**). This is due to the fact that bilayers in decane have smaller areas and larger thicknesses, and, therefore, have smaller capacitance values

wave voltages to estimate values (Table 1) for the equivalent damping and stiffness terms in Eqs. 3, 4 and, for decane, assess the relative contributions of EW and EC on $Q$-$v$ shape. Estimates of $k_{ew}$ (units of tension/length) for bilayers in hexadecane are significantly higher than those in decane, confirming that bilayers in decane are more compliant to EW forces. The lower mean values of EW time constants, $\tau_{ew} = \xi_{ew}/k_{ew}$, for decane also confirm that EW is faster for the smaller, less viscous alkane. The values of the EC-fitted parameters in Table 1 highlight the fact that EC results in slower dynamics and greater resistance to changes in membrane geometry (thickness versus area) compared to EW. Moreover, we found that the damping and stiffness parameters for EW and EC varied with frequency. From this, we surmise this variation in measured materials properties stems from the interfacial membrane not being a closed system with constant volume, since the amount of oil trapped within the

hydrophobic core of the membrane at steady state changes with the driving frequency. This variation is expected to yield differences in the measured material properties. Similar phenomena have been observed for squeeze-film damping systems, where both damping and stiffness parameters are functions of the spacing between the plates[55].

With these values, we simulated responses of the membrane to sinusoidal applied voltages (Fig. 6–8 and Supplementary Figs. 12–19) and computed the percent contributions of EW (changes in $R$) and EC (changes in $W$) to dynamic capacitance at steady state as functions of excitation frequency for decane (Fig. 5a). These comparisons reveal that: (1) EW contributes more than 80% of the change in $C$ at all frequencies; and (2) the contributions of EC, which has a larger time constant, diminish with increasing frequency. As a result, we conclude that EW is the dominant mechanism for changing capacitance (and thus hysteresis) for

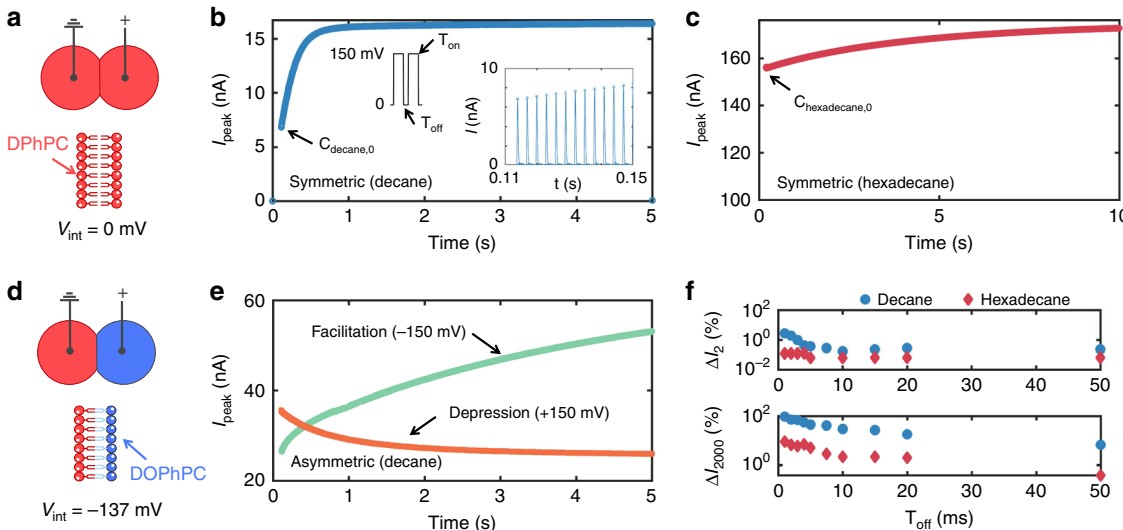

**Fig. 5** Short-term synaptic facilitation and depression achieved via reconfigurable capacitor geometry. **a** A schematic showing a symmetric bilayer with zero intrinsic bias consisting of DPhPC lipids in both leaflets. **b** Demonstration of short-term facilitation of capacitive current measured for a symmetric DPhPC bilayer in decane at RT. Facilitation and depression are forms of short-term plasticity found in the presynaptic terminal[54] evoked by successive voltage stimuli. For this measurement, **a** series of 150 mV pulses with 2 ms ON-time ($T_{on}$) and 0 mV, 1 ms OFF-time ($T_{off}$) caused an accumulated, monotonic increase in peak capacitive current output due to the dynamic, history-dependent increase in membrane capacitance. The inset shows individual capacitive spikes induced by the rising edges of voltage pulses (we are not displaying the negative spikes; Supplementary Fig. 13 provides bipolar responses). **c** Peak capacitive current versus time shows facilitation for a bilayer in hexadecane at RT for the same voltage input as in (**b**). **d** An asymmetric membrane with an intrinsic bias of −137 mV due to the difference in lipid dipole potentials is created by joining one DPhPC-coated droplet and one DOPhPC-coated droplet. **e** Demonstration of short-term depression in peak capacitive current due to a reduction in net membrane potential in response to the same voltage pulses as in (**b**). Changing to negative pulses increases the net membrane potential ($v_m = -v - 137$ mV), which leads to EW and EC increasing the capacitance of the interface, and results in facilitation in peak current versus time. **f** Percentage increases in measured peak current relative to the first pulse response for the 2nd ($\Delta I_2$) and 2000th ($\Delta I_{2000}$) voltage pulses at various $T_{off}$ values for a symmetric DPhPC membrane in decane at RT

**Table 1 Damping and stiffness parameters for decane ($C_{10}$) and hexadecane ($C_{16}$)**

| Oil | Freq. (Hz) | $\zeta_{ew}$ (N s m$^{-2}$) | $k_{ew}$ (N m$^{-2}$) | $\zeta_{ec}$ (N s m$^{-1}$) | $k_{ec}$ (N m$^{-1}$) | $\tau_{ew}$ (s) | $\tau_{ec}$ (s) | MAPE$^a$ (%) | $R^{2a}$ |
|-----|-----------|------|------|------|------|------|------|------|------|
| $C_{10}$ | 0.017 | 0.40 ± .01 | 1.95 ± .02 | 6.45 ± .05 × 10$^6$ | 2.89 ± .004 × 10$^5$ | 0.21 | 22.34 | 3.13 | 0.95 |
| | 0.17 | 0.70 ± .01 | 1.99 ± .01 | 3.54 ± .13 × 10$^6$ | 3.18 ± .03 × 10$^5$ | 0.35 | 11.15 | 2.60 | 0.96 |
| | 0.417 | 0.66 ± .04 | 1.40 ± .01 | 1.99 ± .08 × 10$^6$ | 3.79 ± .05 × 10$^5$ | 0.47 | 5.27 | 1.70 | 0.97 |
| $C_{16}$ | 0.017 | 8.14 ± .03 | 4.42 ± .02 | — | — | 1.84 | — | 1.20 | 0.91 |
| | 0.17 | 4.98 ± .02 | 6.32 ± .01 | — | — | 0.79 | — | 0.33 | 0.98 |
| | 0.417 | 6.33 ± .12 | 11.07 ± .06 | — | — | 0.57 | — | 0.26 | 0.98 |

$^a$See Supplementary Methods for full description of MAPE and $R^2$

DPhPC bilayers, and that bilayers in decane exhibit greater changes in capacitance, more pronounced nonlinearity (Supplementary Fig. 13), and greater $Q$-$v$ hysteresis than hexadecane due to greater compliance of bilayers in decane to EW and the reduction of bilayer thickness due to EC, which is negligible for hexadecane.

Figures 6 and 8b show simulated $R(t)$, $W(t)$, and $Q$-$v$ relationships (at steady state), respectively, induced by sinusoidal voltages for DPhPC bilayers in hexadecane and decane, proving the model captures the frequency dependence of this system. Further, the model can predict the dynamic responses to rectangular voltage pulses, such as those measured experimentally for Fig. 5. Figure 8c shows that the minor axis radius of a bilayer in decane grows quickly by ~35%, accompanied by a slower, smaller (~20%) decrease in thickness. Based on these geometrical reconfigurations of the capacitance, the model also predicts facilitation and depression, respectively, of the peak currents for both symmetric and asymmetric membranes, with the latter simulated by replacing $v(t)$ with $v(t)$−137 mV in Eqs. 3, 4 (Fig. 8d).

## Discussion

Using a soft, modular, and biomimetic membrane that mimics the composition and structure of cellular membranes, we demonstrate volatile memcapacitance via geometrical reconfigurability. Memcapacitive and nonlinear dynamical behaviours, including capacitive short-term facilitation and depression, are found to be governed by two implicitly-coupled, voltage-dependent state variables: membrane radius, $R$, and hydrophobic thickness, $W$. When the bilayer contained less residual oil (e.g., hexadecane), electrowetting is the sole mechanism for variable capacitance (i.e., resulting in a first-order memcapacitor). But when more oil was retained between leaflets (decane) and when this oil had a lower viscosity, both EW and EC contribute to the total change in capacitance, causing $Q$-$v$ to exhibit stronger nonlinearity and reconfigure more quickly to changes in voltage (the decane system is considered a second-order memcapacitor).

In addition to their possible implementation as capacitive synaptic mimics[56] in spike recurrent neural networks for online learning and computation, we envision this system to have impact

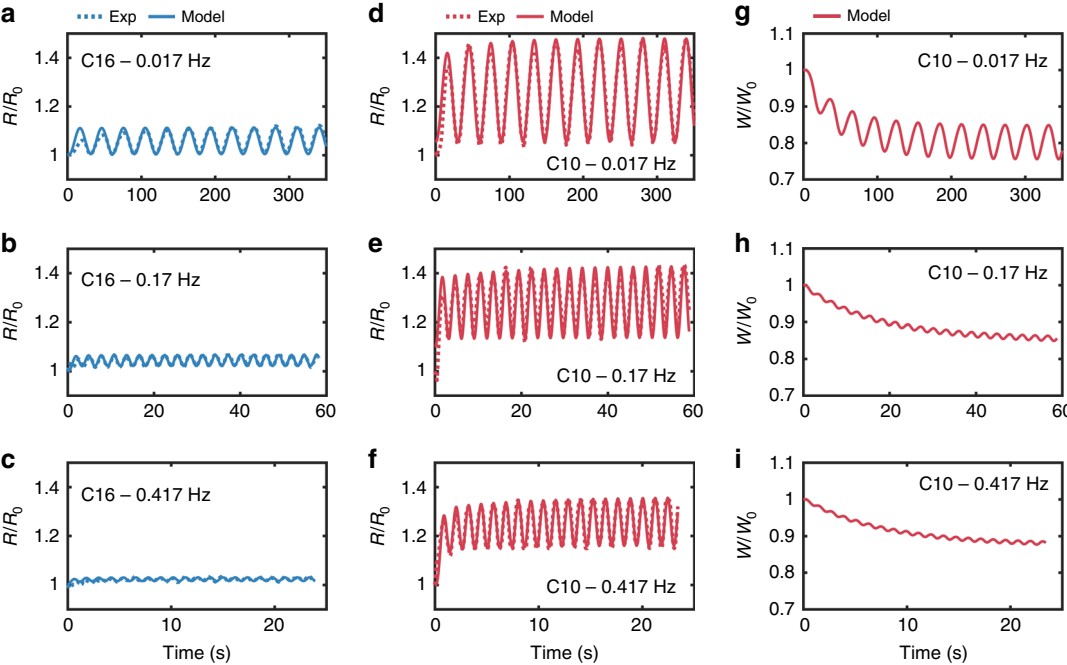

**Fig. 6** Experimental and simulated changes in bilayer radius and thickness in response to sinusoidal voltage input. The first two columns show measurements and simulated (obtained using Eq. 3) responses of dynamic changes in the radii of DPhPC membranes formed in both hexadecane (**a**–**c**) and decane (**d**–**f**) systems in response to sinusoidal voltages at three different frequencies. The third column (**g**–**i**) shows simulated (obtained using Eq. 4) dynamic changes in membrane thickness for a DPhPC membrane formed in decane. Experimental values of dynamic changes in thickness are not obtained; therefore, only simulated results are provided. Also, simulated changes in thickness of DPhPC bilayers in hexadecane are not included since quasi-static measurements show thickness to be constant (Supplementary Fig. 4)

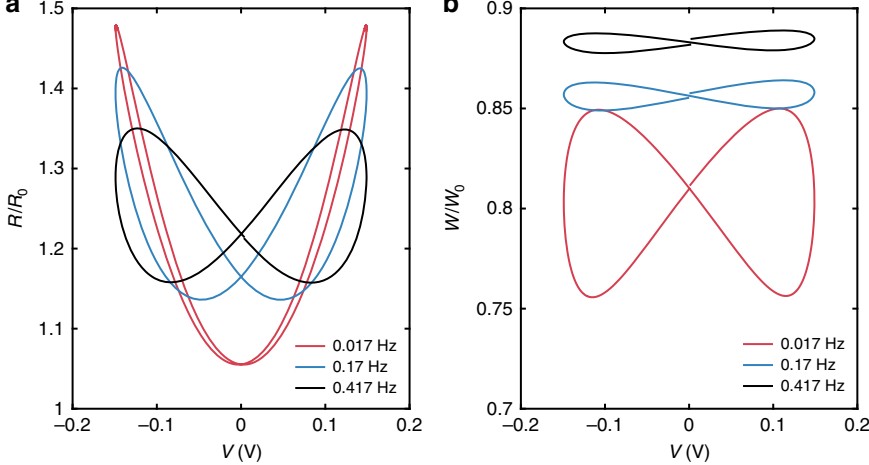

**Fig. 7** Pinched hysteresis in state variable-voltage relationships. Simulations of **a** DPhPC membrane in decane subjected to a sinusoidal voltage show steady-state, pinched hysteresis in both bilayer radius (**a**) and thickness, (**b**). At the three frequencies, the results show that both state variables pinch at zero volts and that the amplitude of change for each state variable decreases as frequency increases. Membrane thickness, which exhibits slower dynamics compared to radius, exhibited greater sensitivity to frequency. These results highlight that the model enables tracking the contribution of each state variable to capacitive memory in the system

in two major areas: (1) modular, low-power materials that could interface with cells and tissues for biosensing and processing of biological signals due to their soft and biocompatible nature[57]; and (2) as a model system to study capacitive excitability in neuronal membranes. The current system differs from biological membranes by having residual oil in the hydrophobic core of the membrane. However, choosing a higher molecular weight oil and naturally-derived lipids could yield more biologically-relevant interfaces with specific capacitances closer to those of biological neurons. We have previously found that membranes from natural

sources such as porcine brain total lipid extract, which contains hundreds of distinct lipid types and large amounts of cholesterol, undergo entropic, higher-order phase transitions detectable with capacitance[40]. Higher-order transitions such as these are believed to be important in cell signalling and dynamic lipid domain (i.e., raft) formation, based on high-amplitude yet nanoscopic and fleeting stochastic fluctuations located near critical points, instead of abrupt changes in molecular composition or density inherent to first-order enthalpic transitions, which tend to be more disruptive to membranes. In the presence of electric fields, charges

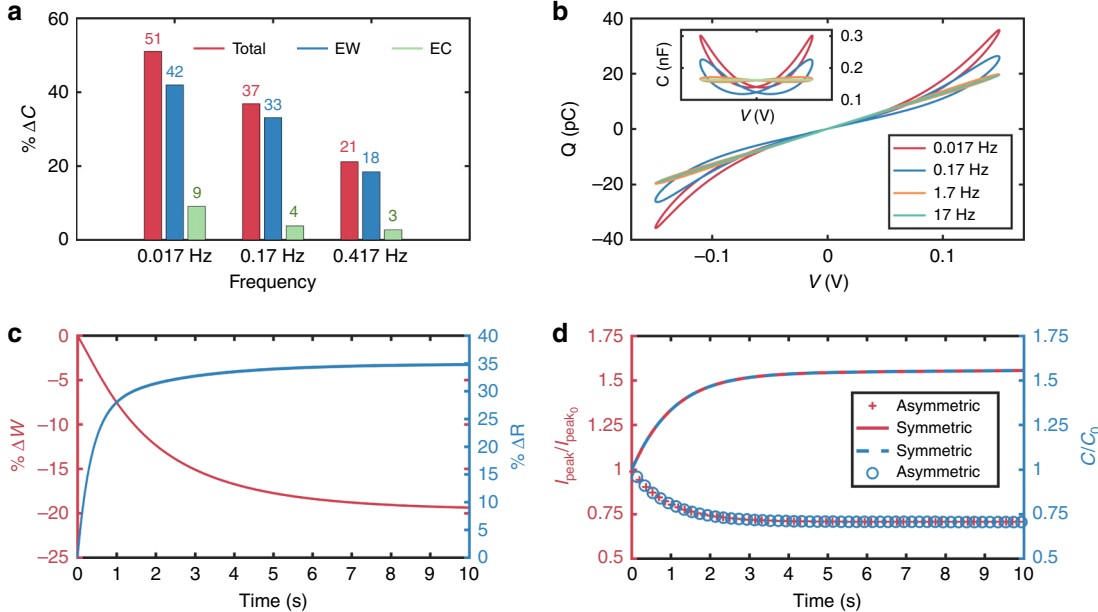

**Fig. 8** Simulated memcapacitance in symmetric and asymmetric membranes in decane. **a** Percentage contributions of EW and EC mechanisms to the steady-state change in capacitance versus excitation frequency. Percentages were computed from the outputs of simulating the geometrical and total capacitive responses of a DPhPC bilayer in decane (Fig. 6). **b** Simulated Q-v loops using the state equations and fitted parameters provided in Table 1. These traces capture the Q-v shape and frequency dependence observed in our experimental results (see Table 1 for MAPE and $R^2$ values) (Fig. 2). **c** Simulation showing the ~20% cumulative decrease in membrane thickness (red) and ~35% increase in membrane radius in response to a rectangular voltage pulse train (150 mV, $T_{on} = 2$ ms, $T_{off} = 1$ ms). **d** Simulated peak currents (red and green, respectively) and capacitance (blue) responses corresponding to the symmetric DPhPC membrane in decane described in (**b**) and (**c**) and the asymmetric DPhPC:DOPhPC membrane in decane. The model captures both forms of short-term plasticity observed in Fig. 5

exert forces on the membrane that can change its molecular ordering via electrostriction (EW and EC). This results in a capacitive susceptibility that replaces the common assumption of constant capacitance, which, up to now, has dominated electrophysiological descriptions and characterizations of biomembranes. Such experiments could bring completely new insights regarding the capacitive susceptibility of neuronal membranes, and therefore, may impact the Hodgkin-Huxley model for excitability, which assumes capacitance to be a constant.

## Methods

**Preparation of lipid solutions and membrane assembly.** The aqueous droplets consist of deionized water (18.2 MΩ.cm) containing 2 mg ml$^{-1}$ of either DPhPC or DOPhPC liposomes, 500 mM potassium chloride (KCl, Sigma), and 10 mM 3-(N-morpholino)propanesulfonic acid (MOPS, Sigma), with a pH of 7.0. Both DPhPC and DOPhPC liposomes vesicles are prepared and stored as described elsewhere[37]. To help the droplets anchor to the silver/silver chloride (Ag/AgCl) wires (Goodfellow), we coat their ball-ended tips with a 1% agarose gel solution. The oil surrounding the droplets consists of either decane (≥95%, Sigma) or hexadecane (≥99%, Sigma). Lipid membranes are formed between two aqueous droplets anchored to wire-type electrodes in a transparent reservoir filled with oil, as described elsewhere[16,37,41].

**Electrical measurements.** To record the lipid membrane formation, which is reflected as an increase in membrane capacitance, we supply a 10 Hz, 10 mV triangular wave to the electrodes using a function generator. Due to the capacitive nature of the membrane, the resulting current response is square-like (Supplementary Fig. 2). As the area of the thinned lipid membrane grows, the peak-to-peak current amplitude increases until reaching a steady state (Supplementary Fig. 2). To obtain the C-v and Q-v plots, we use a custom LabView code to apply a low-frequency, high-amplitude sinusoidal voltage waveform (amplitude and frequencies are mentioned in the main text) overlaid with a 10 mV, 100 Hz triangular voltage waveform output from an Agilent 33210A waveform generator. While the low-frequency, sinusoidal waveform drives geometrical changes at the lipid interface (i.e., EW and EC), the capacitance measurements are based on the lipid membrane's current response to the higher frequency triangular waveform. The capacitance of the lipid interface is then extracted from sections of the square-wave current response using a custom MATLAB script (available upon request). In

parallel, to monitor the changes in the membrane's minor axis radius, R, we acquire images of the droplets as viewed from below through a 4x objective lens on an Olympus IX51 inverted microscope using a QI Click CCD. We then post-process the images using custom scripts in MATLAB to extract values of R. For the signal pulse experiments, we use a custom LabView code to generate pulses with specific amplitudes and ON and OFF times. All current recordings are made using an Axopatch 200B patch clamp amplifier and Digidata 1440 data acquisition system (Molecular Devices). For all measurements, droplets and measurement probes are placed under a lab-made Faraday cage to minimize noise from the environment. To obtain the energy dissipation displayed in Fig. 3, we integrate with respect to time the product of applied voltage bias, $v(t)$, and capacitive current, $I_c$, given by

$$I_c = C\frac{dv(t)}{dt} + v(t)\frac{dC}{dt}, \qquad (5)$$

where $C$ the memcapacitance of the lipid membrane and $v(t)$ is the voltage bias.

## Data availability
All relevant data that support the findings of this study are available from the corresponding authors on reasonable request.

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

## Acknowledgements

Financial support was provided by the National Science Foundation Grant NSF ECCS-1631472. A portion of this research was conducted at the Center for Nanophase Materials Sciences, which is a DOE Office of Science User Facility. The graphics in Figs. 1, 5 and Supplementary Fig. 1 were modified from Servier Medical Art (http://smart.servier.com), licensed under a Creative Common Attribution 3.0 Generic License (https://creativecommons.org/licenses/by/3.0/). This manuscript has been authored by UT-Battelle, LLC, under Contract No. DE-AC0500OR22725 with the U.S. Department of Energy. The United States Government retains and the publisher, by accepting the article for publication, acknowledges that the United States Government retains a non-exclusive, paid-up, irrevocable, world-wide license to publish or reproduce the published form of this manuscript, or allow others to do so, for the United States Government purposes.

## Author contributions

J.S.N., S.A.S., R.S.W. and C.P.C. designed experiments; J.S.N. performed experiments with input from S.A.S., R.S.W., G.J.T. and C.P.C.; J.S.N. analyzed data with input from S. A.S., R.S.W., M.S.H. and C.P.C.; J.S.N. and M.S.H. developed mathematical models with input from R.S.W., S.A.S. and C.P.C.; M.S.H. performed the data fitting and model simulations with inputs from J.S.N., R.S.W., R.J.W. and G.S.R.; J.S.N., M.S.H., S.A.S., R.S. W. and C.P.C. wrote the paper; J.S.N. prepared figures and videos; J.S.N. and M.S.H. prepared Supplementary Information; J.S.N., R.S.W., S.A.S. and C.P.C conceived the concept; S.A.S. and C.P.C. managed and supervised the project. All discussed the results and commented on the manuscript at all stages.

## Additional information

**Competing interests:** The authors declare no competing interests.

