## [Peer Review File · Nature Communications]

Reviewers' comments:

Reviewer #1 (Remarks to the Author):

This is an interesting manuscript that addresses memcapacitive effects in synthetic biomembranes. Using experiments and a model that takes into account voltage-controlled geometrical changes of a lipid bilayer (through electrowetting and electrocompression), the authors were able to link geometrical changes and inertia of the membrane with its volatile memory phenomena. They also describe a link between membrane physics and memcapacitance phenomena by expressing the capacitance of the membrane as a function of its internal state variables (thickness and area of the membrane). The manuscript is densely written, well organized and the topic is interdisciplinary. Therefore I believe that this manuscript suits to the purpose of the journal and it could be a nice contribution to Nature Communications. Nevertheless, I believe that the authors should address the following issues in order to improve the readability and the quality of the manuscript:

1. At the introduction, the authors refer to the limitations of 3-terminal devices (scaling limitations, inherent limit due to Moore's law) and the advantages of 2-terminal devices in certain aspects (integration density). Although the proposed structure adheres to the generic 2-terminal structure and is very interesting for basic research, the authors give no clue for simple functional circuits. The authors could add a comment on this part.
2. At the introduction the authors state "devices in which memcapacitance originates from geometrical changes remain unrealized to this point". The authors could give examples of memcapacitive mechanical oscillators (M. Di Ventra, Y. Pershin, *Advances in Physics*, 60, 2011). The authors could also cite previous relevant works on memcapacitive effects in nanopores (M. Krems, Y. Pershin, M. Di Ventra, *Nano Lett.* 10 (7), 2674–2678, 2010).
3. Is there any evidence in simulations or experimental measurements of negative capacitance effects under certain conditions (for example in other frequency regimes)?
4. In their model, the authors assume that the lipids are totally immobile. Is this totally realistic? If not, are there any consequences at the electrical behaviour of the bi-layer (in higher frequencies for example, where usually polarization phenomena appear)?
5. In the previous work, the authors incorporated voltage gated channels (alm peptides) at the bilayer, producing thus a diode-like behavior. What is the difference between the two works? In the previous work, the authors assumed an interfacial active area or thickness which was independent on the external electrical perturbation? Which approach is closer to reality?

Reviewer #2 (Remarks to the Author):

The authors construct a bilayer interface between two water droplets in oil, and apply a sine waveform across the interface by inserting two electrodes in each water droplet containing electrolyte. This is basically a measurement of electrical double layer effect (of lipid bilayer), charging and discharging process. Although this is the kind of problem studied for decades by biophysics and electrochemistry community, there are merits worth consideration for Nat. Comm.

The memory effect and the perceived applications are intriguing, but there are basic questions to be deliberated.

An essential question is the concepts of transport charges and hysteresis charges are not clearly defined or differentiated (fig 3 & related). This is a core to the whole presented picture. The authors should revisit their data analysis and discussion in reference to recent literature such as (ChemElectroChem 2018, 5 (20), 3089) and related. The energy dissipation term appears to include multiple convoluted sources with different time constants. Related, it is an overstatement to claim the "first example of voltage-controlled memcapacitor".

The very interesting section is when they break the symmetry of the interface by using two types of lipids. It is unclear whether the responses display broken symmetry across the V sub int. I suggest them to extend the V range in Fig. S 10 or add a DC bias. My prediction is that they might have a diode type system here.

Overall, a very interesting work that deserves consideration.

Reviewer #3 (Remarks to the Author):

This paper discusses the experimental realization and the theoretical description of a memcapacitor realized by biomimetic membranes.

The paper is well written and the experiments are well conceived. However, I have a few concerns about it that make me think the paper is not appropriate for Nature Comm., but rather for another journal with important revisions.

First of all, the authors acknowledge that memcapacitors have already been reported in the literature. Even the one reported here has been already discussed by the same authors in their previous work (Ref. 16 of this paper), although not fully developed as in the present paper.

The authors then claim that all previous results, except one, "failed to provide comprehensive evidence of a pinched hysteresis in the charge-voltage plane as proof of ideal memory capacitance".

The authors here make a fundamental mistake and propagate a misunderstanding: it is not true that a pinched hysteresis loop is "proof of memory capacitance". In fact, a non-crossing loop is still a valid memcapacitive system as it is for any other memory element. The reason is that the capacitance (like the resistance) is a derived quantity (a response function), therefore it can take any value, including negative and infinite one, while the voltage and the charge are always finite (see, e.g., Nanotechnology, 24, 255201, 2013).

In other words, unlike what the authors claim, all previous reports of memcapacitors are all valid ones.

This leads me to the issue of novelty. This paper is an interesting addition to the literature of memory elements, but it is far from being novel or groundbreaking: there are already other papers out there that discuss memcapacitors (including suggestions from the authors' Ref. 16 using the same system), and this one does not seem to be such an outstanding advancement to warrant publication in Nature Comm.

Also the vague statement "These findings will accelerate the development of low-energy, biomolecular neuromorphic memelements" is quite weak and not supported.

Therefore, I strongly suggest the authors to submit this paper to another journal by changing the

above claim on the pinched input-output curve and providing the correct acknowledgment of the previous papers on memcapacitors.

Reviewer #1 (Remarks to the Author):

This is an interesting manuscript that addresses memcapacitive effects in synthetic biomembranes. Using experiments and a model that takes into account voltage-controlled geometrical changes of a lipid bilayer (through electrowetting and electrocompression), the authors were able to link geometrical changes and inertia of the membrane with its volatile memory phenomena. They also describe a link between membrane physics and memcapacitance phenomena by expressing the capacitance of the membrane as a function of its internal state variables (thickness and area of the membrane). The manuscript is densely written, well organized and the topic is interdisciplinary. Therefore I believe that this manuscript suits to the purpose of the journal and it could be a nice contribution to Nature Communications. Nevertheless, I believe that the authors should address the following issues in order to improve the readability and the quality of the manuscript:

1. At the introduction, the authors refer to the limitations of 3-terminal devices (scaling limitations, inherent limit due to Moore's law) and the advantages of 2-terminal devices in certain aspects (integration density). Although the proposed structure adheres to the generic 2-terminal structure and is very interesting for basic research, the authors give no clue for simple functional circuits. The authors could add a comment on this part.

In this manuscript, we demonstrated with both experiments and simulations that our two-terminal, *lipid-only* (i.e., no conductive ion channels/pores present within the membrane) memcapacitor exhibits facilitation and depression—two types of short-term plasticity exhibited in the presynaptic terminal of biological synapses. We envision circuits that would implement this dynamic system as a *memcapacitive synapse* for: 1) online learning in integrate-and-fire neural networks and 2) adaptive signal processing. These applications are analogous to those we described recently^{1,2} on the implementation of two-terminal, *memristive synapses* constructed from biomembranes containing voltage-activated ion channels (nanopores). In these publications, we showed through both simulations and experiments that *resistive synapses* could be connected to solid-state neurons in spiking, recurrent neural networks for online learning applications.

Changes to the manuscript

To provide this additional context, we have added references to our prior work implementing resistive synapses in the Introduction (lines 81-83), and a statement regarding potential implementation in brain-inspired circuits in the Conclusion (lines 345-346).

2. At the introduction the authors state “devices in which memcapacitance originates from geometrical changes remain unrealized to this point”. The authors could give examples of memcapacitive mechanical oscillators (M. Di Ventra, Y. Pershin, *Advances in Physics*, 60, 2011). The authors could also cite previous relevant works on memcapacitive effects in nanopores (M. Krems, Y. Pershin, M. Di Ventra, *Nano Lett.* 10 (7), 2674–2678, 2010).

1. The first suggested reference is a review article that discusses memcapacitive systems and includes citations to multiple examples of micro-electromechanical system (MEMS) devices exhibiting voltage-controlled capacitance. These show that it is possible to design/achieve

hysteretic capacitance in bistable, mechanical oscillators (e.g., microscale dielectric beams) that “snap” between buckled states of deflection when excited by an electric field. The bistable deformation states engineered through top-down fabrication methods result in *discrete*, non-volatile, and hysteretic changes in measured capacitance versus voltage, demonstrating memory capacitance.

2. The second reference suggested by the reviewer is a theoretical study of memory capacitance in silicon-nitride membranes containing nanoscale (0.7 nm diameter) pores, where the observed “unusual capacitance” exhibited a non-pinned hysteretic charge-voltage relationship, which does not meet the requirement for an ideal memory capacitor as defined by Chua. Instead, variations in capacitance were likely due to non-capacitive, ionic charge leakage through the pores in the membrane. In our response to reviewer 3 below, we describe why a non-pinned charge-voltage plot does not satisfy the requirements of an ideal memory capacitor.

Changes to the manuscript

- Based on the reviewer’s comments, we have referenced the review article in line 71 of the manuscript where we list prior examples of physically-realized memory capacitors.
- Further, to more accurately describe the significance of our own work, we have changed our statements on lines 72-77 to read, “Of these, most did not provide evidence of pinched hysteresis in the charge-voltage plane as proof of memory capacitance, as first defined by Chua^{14,32}, or develop realistic, physics-based models to describe and predict the state variables driving capacitive reconfigurations. Devices for which ideal, analogue memcapacitance that originates from geometrical changes in materials *at the molecular scale* remain unrealized to date.”

3. Is there any evidence in simulations or experimental measurements of negative capacitance effects under certain conditions (for example in other frequency regimes)?

We have not observed negative capacitance in any of our measurements at the conditions specified in the manuscript and SI, though we have observed signatures of negative differential resistance (NDR) in some memristive devices (separate paper in progress) at similar excitation frequencies.

Changes to the manuscript

No changes made to the manuscript or SI.

4. In their model, the authors assume that the lipids are totally immobile. Is this totally realistic? If not, are there any consequences at the electrical behaviour of the bi-layer (in higher frequencies for example, where usually polarization phenomena appear)?

Our model makes no assumption that the lipids are immobile. Rather, the model simply assumes that the average material properties and composition of each lipid leaflet of the bilayer do not change at constant temperature. This reflects the fact that it should be energetically unfavorable for the density of lipids in the leaflets to spontaneously change or for lipids in asymmetric membranes to spontaneously flip to the other leaflet, which could change the intrinsic membrane

potential. We have shown that lipid adsorption at an oil-water interface quickly reaches an equilibrium density^{3,4}, and that lipid flip-flop in asymmetric, lipid-only membranes is very slow⁵. Therefore, we do not expect local changes in the composition or density of the monolayers that form bilayer membranes to affect the macroscale capacitance of the entire interface. Possible effects that lipid mobility or polarizability would have at much higher frequencies were not considered given the very low frequencies (<2 Hz) where memory capacitance was observed.

Changes to the manuscript

No changes made to the manuscript or SI.

5. In the previous work, the authors incorporated voltage gated channels (alm peptides) at the bilayer, producing thus a diode-like behavior. What is the difference between the two works? In the previous work, the authors assumed an interfacial active area or thickness which was independent on the external electrical perturbation? Which approach is closer to reality?

The key difference in this study versus our prior work published in ACS Nano⁶ (2018) is that the membranes assembled and characterized here contain **no ion channels/pores in the membrane**. These **lipid-only membranes** are thus highly insulating and well-described as capacitors.

Changes to the manuscript

Changed “*void of*” to “*without*” on line 84 to emphasize that the membranes studied here do not contain ion channels or pores. We have also added a statement that further clarifies that charges do not translocate across the membrane in our system (lines 119-120).

Reviewer #2 (Remarks to the Author):

The authors construct a bilayer interface between two water droplets in oil, and apply a sine waveform across the interface by inserting two electrodes in each water droplet containing electrolyte. This is basically a measurement of electrical double layer effect (of lipid bilayer), charging and discharging process. Although this is the kind of problem studied for decades by biophysics and electrochemistry community, there are merits worth consideration for Nat. Comm.

The memory effect and the perceived applications are intriguing, but there are basic questions to be deliberated. An essential question is the concepts of transport charges and hysteresis charges are not clearly defined or differentiated (fig 3 & related). This is a core to the whole presented picture. The authors should revisit their data analysis and discussion in reference to recent literature such as (ChemElectroChem 2018, 5 (20), 3089) and related.

The energy dissipation term appears to include multiple convoluted sources with different time constants. Related, it is an overstatement to claim the "first example of voltage-controlled memcapacitor".

It appears that the reviewer's question assumes that we have conductive pores in the membrane, which is not the case in the present study. As emphasized in the reply to **Reviewer 1** above, the membrane systems here are highly insulating, lipid-only bilayer membranes. Therefore, there should be no transport of ions across the interface. Instead, the charge versus voltage relationships reflect only capacitive behaviours. As explained in our model and data analysis, the two different time constants stem from the two different mechanisms—electrowetting and electrocompression—for altering the geometry of the capacitor.

Regarding the concern about the phrase, "the first example of a voltage-controlled memcapacitor" being an overstatement, we want to make sure that this phrase was taken in the context of the entire sentence, which reads, "Here, we report the first example of a voltage-controlled memcapacitor in which activity-dependent capacitive plasticity arises from reversible and hysteretic geometrical changes in a lipid bilayer that mimics the composition and structure of biomembranes." To our knowledge, no study has been published to date describing capacitive memory in synthetic biomembranes.

Changes to the manuscript

No changes made to the manuscript or SI.

The very interesting section is when they break the symmetry of the interface by using two types of lipids. It is unclear whether the responses display broken symmetry across the V sub int. I suggest them to extend the V range in Fig. S 10 or add a DC bias. My prediction is that they might have a diode type system here.

We demonstrated that a biomimetic membrane consisting of two leaflets with different lipid compositions (i.e., with different dipole potentials) results in a nonzero intrinsic membrane voltage ($V_{int} \sim -140$ mV). In this case, the net membrane voltage, which is the driving force for electrowetting and electrocompression, is $V_m(t) = V_{bias}(t) + V_{int}$ for an applied voltage $V_{bias}(t)$. Thus, for a symmetric, sinusoidal $V_{bias}(t)$, the membrane will exhibit changes in geometry (i.e., capacitance) asymmetrically with respect to $V_{bias}(t)$, but symmetrically with respect to $V_m(t)$. This results in a hysteretic charge-voltage loop that pinches at $V_{bias}(t) = -V_{int}$. The results provided in Figure S10 of the SI demonstrate this asymmetric pinching (and diode-like response) for a symmetric applied potential. To visualize the response across a larger range of applied potentials, we show here simulated $Q - V$ responses at a higher range of bias voltages ($|V_{bias}| = 500$ mV) compared to the bias range shown in Figure S10 ($|V_{bias}| = 190$ mV).

On another note, a diode has an asymmetric response in the $i-v$ plane that can be observed using quasi-DC measurements. Our device has an asymmetric response in the $q-v$ plane that can only be observed for an AC excitation.

Changes to the manuscript & SI

This figure and a detailed caption have been added as Figure S11 in the SI, and this figure has been cross-referenced on line 254. We have also updated the cross-references for Figs. S12, S13, and S14 on lines 275, 305, 311, and 326 in the manuscript.

Overall, a very interesting work that deserves consideration.

Reviewer #3 (Remarks to the Author):

This paper discusses the experimental realization and the theoretical description of a memcapacitor realized by biomimetic membranes.

The paper is well written and the experiments are well conceived. However, I have a few concerns about it that make me think the paper is not appropriate for Nature Comm., but rather for another journal with important revisions.

First of all, the authors acknowledge that memcapacitors have already been reported in the literature. Even the one reported here has been already discussed by the same authors in their previous work (Ref. 16 of this paper), although not fully developed as in the present paper.

The authors then claim that all previous results, except one, “failed to provide comprehensive evidence of a pinched hysteresis in the charge-voltage plane as proof of ideal memory capacitance”.

The authors here make a fundamental mistake and propagate a misunderstanding: it is not true that a pinched hysteresis loop is “proof of memory capacitance”. In fact, a non-crossing loop is still a valid memcapacitive system as it is for any other memory element. The reason is that the capacitance (like the resistance) is a derived quantity (a response function), therefore it can take any value, including negative and infinite one, while the voltage and the charge are always finite (see, e.g., *Nanotechnology*, 24, 255201, 2013).

In other words, unlike what the authors claim, all previous reports of memcapacitors are all valid ones.

This leads me to the issue of novelty. This paper is an interesting addition to the literature of memory elements, but it is far from being novel or groundbreaking: there are already other papers out there that discuss memcapacitors (including suggestions from the authors’ Ref. 16 using the same system), and this one does not seem to be such an outstanding advancement to warrant publication in Nature Comm.

Also the vague statement “These findings will accelerate the development of low-energy, biomolecular neuromorphic memelements” is quite weak and not supported.

Therefore, I strongly suggest the authors to submit this paper to another journal by changing the above claim on the pinched input-output curve and providing the correct acknowledgment of the previous papers on memcapacitors.

The paper cited by the reviewer has been thoroughly rebutted by Prof. Leon Chua in his tutorial “If it’s pinched, it’s a memristor,” [*Semicond. Sci. Technol.* **29**, 104001 (2014)]. The misconception and propagation of misunderstanding is in the *Nanotechnology* paper. The relevant discussion from Chua’s tutorial is shown below in its entirety. What the authors of

the Nanotechnology paper failed to grasp is that the *definitions* of memristor, memcapacitor and other fundamental circuit elements are purely mathematical, independent of *any* physical mechanism. They are basic elements that describe a consistent and independent set of ideal behaviours. In order to ensure the independence of these ideal circuit elements, they must adhere to strict mathematical constraints, one of which is that the property of the device being modelled (e.g. resistance or capacitance) must be bounded. In the material below, extracted from his tutorial, Chua shows that if this constraint is not obeyed, then one obtains obvious contradictions and anomalies, such that a memristor is a capacitor or vice versa. Clearly, such ambiguity makes the resulting model useless.

No physical device is perfectly described by Chua's fundamental circuit elements, but an accurate and useful device model will normally involve an equivalent circuit in which two or more of the elements are utilized. This has been described in detail by Chua nearly 40 years ago ["Device Modelling Via Basic Nonlinear Circuit Elements," *IEEE Trans Circuits Systems CAS* **27**,1014-1044 (1980)]. Thus, when introducing dissipation (no matter the source) into a model for a physical device that has a capacitance, one does not alter the fundamental definition of a capacitor, but rather introduces an appropriate resistor in parallel with a capacitor to include the leaky or dissipative nature of the physical device. This is the origin of another major misconception in the Nanotechnology paper. The authors describe what they claim to be a memcapacitor, but in fact it is not a memcapacitor at all but a linear capacitor with a parallel nonlinear resistor (i.e., a leaky capacitor). This is illustrated clearly below by comparing Figure 1 from the Nanotechnology paper to a simple calculation for an ideal linear capacitor in parallel with a linear resistor driven by a sinusoidal voltage. The presence of the dissipative element opens a hysteresis loop in the charge vs. voltage plot for the model leaky capacitor. This happens because the resistance introduces a phase shift between the charge and voltage. The rather astounding mistake in the Nanotechnology papers occurs when the authors claim that the capacitance of their model device is a function of voltage and plot the AC charge divided by the AC voltage vs. AC voltage to obtain a plot that diverges at zero voltage! A very similar plot is obtained for the linear leaky capacitor, for which the capacitance is a constant, and thus obviously neither voltage dependent nor divergent. The authors of the Nanotechnology paper apparently fail to understand that modelling an AC circuit requires explicit knowledge of the phase shifts between voltages, currents and charges. We thus see that the entire concept of the Nanotechnology paper was fundamentally incorrect.

To conclude, it is true that for any physical realization, a memcapacitor must have a pinched hysteresis loop – that is a necessary consequence of the definition. What is interesting here is that we observe a system that displays an unambiguous memcapacitance – this has not been so clear in previous works or has in fact been mistakenly claimed. What is intriguing about this observation is that it may be important for signal analysis and transmission in biological systems.

TUTORIAL

If it's pinched it's a memristor

To cite this article: Leon Chua 2014 *Semicond. Sci. Technol.* **29** 104001

9. If it's not pinched it's not a memristor

[Redacted]

[Redacted]

From the paper cited by Referee 3:

Nanotechnology 24 (2013) 255201

Our simulations for an ideal linear capacitor in parallel with a linear resistor driven by a sinusoidal voltage:

[Redacted]

q vs. v for leaky capacitor

C vs. v for leaky capacitor

References:

- 1 Weiss, R. *et al.* in *2018 IEEE Biomedical Circuits and Systems Conference (BioCAS)*. 1-4 (IEEE).
- 2 Hasan, M. S. *et al.* in *2018 IEEE 13th Dallas Circuits and Systems Conference (DCAS)*. 1-6 (IEEE).
- 3 Venkatesan, G. A. *et al.* Evaporation-induced monolayer compression improves droplet interface bilayer formation using unsaturated lipids. *Biomicrofluidics* **12**, 024101, doi:10.1063/1.5016523 (2018).
- 4 Venkatesan, G. A. *et al.* Adsorption Kinetics Dictate Monolayer Self-Assembly for Both Lipid-In and Lipid-Out Approaches to Droplet Interface Bilayer Formation. *Langmuir* **31**, 12883-12893, doi:10.1021/acs.langmuir.5b02293 (2015).
- 5 Taylor, G. *et al.* Electrophysiological interrogation of asymmetric droplet interface bilayers reveals surface-bound alamethicin induces lipid flip-flop. *Biochimica et Biophysica Acta (BBA) - Biomembranes* **in print**, doi:<https://doi.org/10.1016/j.bbamem.2018.07.001> (2018).
- 6 Najem, J. S. *et al.* Memristive Ion Channel-Doped Biomembranes as Synaptic Mimics. *Acs Nano* (2018).

Reviewers' comments:

Reviewer #1 (Remarks to the Author):

The authors addressed most of my comments in an sufficient way. In this form, the paper differentiates from the previous works of the authors, as well as from other existing and relevant reports. I believe that the manuscript is suitable for publication in Nat. Commun.

Reviewer #2 (Remarks to the Author):

I am glad to see the new results indeed reveal rectified responses as I expected. If my following suggestions are unclear, it might help to view their system as two EDLs back-in-back, or liquid-liquid interface mimics (books by Conway on supercapacitors or Bard on electrochemistry on EDL). The authors should add comments on the V sub int in comparison to the transmembrane potential in nanopores (Ref. 24).

The nanopore literature become highly relevant with the new results. Whether the EDL is in a nanopore or across the bilayer, it is the charge transport that is measured/affected. Key references (for example Chem. Sci. 2014, 1827; ChemElectroChem 2018, 5 (20), 3089) are missed which make the (revised) into an understatement of the progresses in the field.

Previous suggested charge analysis requires no assumption of pore existence, nor depends on it. The authors are again suggested to consider the physical origin of their observed behaviors. This in part relates to their rebuttal to reviewer 3 comments. Be it the rigor in definition or introducing multiple elements to build equivalent circuits, the physical process arises from charge transport or redistribution.

The measured current/charge and hysteresis effect in their system, unlike solid state elements, are from anions as the main charge carriers on one side (of the bilayer) while cations the other. Those will vary under AC stimulus. Therefore, the respective contributions at a given time/frequency range can be important to consider. The changes in charges (not transported charges) within a period/cycle (or half cycle), i.e. those within the pinched loops, can be significant to analyze and might correlate with the proposed size/shape (which translates into effective EDL structure/volume) variations. If full charge analysis could not be incorporated, at least they should briefly comment on the diffusion coefficient of individual charge carriers, the time constant of their system, and qualitatively respective contributions within a given representative condition.

To the other note, one can always convert/replot the same results in i-V or Q-V format (meaning is a different story) as long as those were collected under a regular waveforms (AC sine or triangular).

Reviewer #3 (Remarks to the Author):

The authors of this paper did not answer my concerns at all. Instead, they seem to have major misunderstandings about the nature of their work which makes my original assessment even more compelling.

For one, the paper by Chua the authors report in their reply does not address the issues raised by the paper in Nanotechnology, 24, 255201, 2013 since Chua's paper seems completely oblivious to the well-known theory of response functions. By citing Chua's paper the authors themselves seem to be totally ignorant about this fundamental theory.

Second, of course a linear capacitor in parallel to a nonlinear resistor (a leaky capacitor) would show a

non-pinched hysteresis and hence divergent capacitance. That is not the point.

What the Nanotechnology paper I was referring to discusses is that capacitances (like resistances and inductances) are derived quantities. It never discusses capacitors in parallel with resistors. I wonder if these authors have actually read it.

The only measurable quantities (in the case of a capacitor) are charges and voltages. Therefore, when one extracts from these two quantities the capacitance one can get a finite, infinite, negative or positive capacitance. This is true whether one looks at it physically or just "mathematically". There is nothing strange about this and should be well known. Sadly, neither Chua nor these authors seem to know this.

That is all that Nanotechnology paper is discussing. If the authors think this is "fundamentally incorrect" then they really do not understand basic concepts.

The capacitor the authors discuss in their paper is pinched because the memory arises from moving the capacitor plates and not from dielectric permittivity variations. Both are valid sources of memory.

But take for instance a capacitor whose memory comes only from a slow change in dielectric permittivity in response to the voltage (not a leaky capacitor or one with moving plates).

This capacitor will show hysteretic non-pinched q - V curves and diverging capacitance. It is a legitimate "memcapacitor" but not one showing pinched hysteresis. The memory comes from the dielectric permittivity not from a parallel resistor.

Therefore, I stand by my previous comments: there is nothing particularly novel or groundbreaking about the present work to warrant publication in Nature Communications.

There have been other reported cases of memcapacitors as pointed out by the authors. The authors themselves discuss the same concept in their previous publication (Ref. 16 of their paper). This new paper is yet another example of memcapacitor but not one particularly compelling.

In addition, with their reply the authors show a complete lack of understanding of basic concepts. They do not seem to realize this even after my comments.

Due to the lack of novelty and the wrong and misleading statements by the authors I therefore think this paper is not suitable for Nature Communications.

Our rebuttal comments are provided below in blue text. All changes to the manuscript and SI have been noted.

Reviewer #1 (Remarks to the Author):

The authors addressed most of my comments in an sufficient way. In this form, the paper differentiates from the previous works of the authors, as well as from other existing and relevant reports. I believe that the manuscript is suitable for publication in Nat. Commun.

Reviewer #2 (Remarks to the Author):

I am glad to see the new results indeed reveal rectified responses as I expected. If my following suggestions are unclear, it might help to view their system as two EDLs back-in-back, or liquid-liquid interface mimics (books by Conway on supercapacitors or Bard on electrochemistry on EDL). The authors should add comments on the V sub int in comparison to the transmembrane potential in nanopores (Ref. 24).

The nanopore literature become highly relevant with the new results. Whether the EDL is in a nanopore or across the bilayer, it is the charge transport that is measured/affected. Key references (for example Chem. Sci. 2014, 1827; ChemElectroChem 2018, 5 (20), 3089) are missed which make the (revised) intro an understatement of the progresses in the field.

Previous suggested charge analysis requires no assumption of pore existence, nor depends on it. The authors are again suggested to consider the physical origin of their observed behaviors. This in part relates to their rebuttal to reviewer 3 comments. Be it the rigor in definition or introducing multiple elements to build equivalent circuits, the physical process arises from charge transport or redistribution.

The measured current/charge and hysteresis effect in their system, unlike solid state elements, are from anions as the main charge carriers on one side (of the bilayer) while cations the other. Those will vary under AC stimulus. Therefore, the respective contributions at a given time/frequency range can be important to consider. The changes in charges (not transported charges) within a period/cycle (or half cycle), i.e. those within the pinched loops, can be significant to analyze and might correlate with the proposed size/shape (which translates into effective EDL structure/volume) variations. If full charge analysis could not be incorporated, at least they should briefly comment on the diffusion coefficient of individual charge carriers, the time constant of their system, and qualitatively respective contributions within a given representative condition.

To the other note, one can always convert/replot the same results in i - V or Q - V format (meaning is a different story) as long as those were collected under a regular waveforms (AC sine or triangular).

We had hoped that our first rebuttal to these questions and the strong experimental evidence provided in the manuscript, including 1) visual evidence of area changing at similar dynamic rates and consistent magnitudes to changes in capacitance; 2) and control experiments proving that hysteresis vanishes when the membrane is formed in the presence of a high molecular weight oil that prevents electrowetting and electrocompression; and 3) physics based models to describe our sources of memory capacitance, would sufficiently explain the physics that drive dynamic hysteresis in our system. Yet, the reviewer continues to question the possible role of electrical double layer (EDL) charging/discharging dynamics. And after again thoroughly reading the suggested papers^{1,2}, we now clearly understand the nature of this confusion, since these works showed that glass substrates with conical nanopores can exhibit current-voltage hysteresis due to EDL dynamics at similar dynamic rates (time constants ~ 1 s). However, this similarity in dynamic rates appears to be a coincidence, given the following key differences:

- Our system consists of a membrane that can be modeled by a capacitor (~ 500 pF) connected in parallel to a resistor (~ 10 G Ω), which are connected in series to a resistor (~ 10 k Ω) that represents the electrical resistance of the aqueous electrolyte in the droplets. For this equivalent circuit and these relative values of resistance, the time constant for charging/discharging the electrical double layer on the membrane is determined by the product of membrane capacitance—a value that is dominated by

the dielectric core of the membrane (i.e. not the EDL)³—and the electrolyte resistance. Thus, the EDL that form by mobile ions on the surfaces of our planar membranes charge and discharge with a time constant of $\sim 5\mu\text{s}$. This time constant suggests that hysteresis due to ion rearrangement would appear at frequencies close to 20kHz, far beyond those for which we observed hysteresis in membrane charging. This is even higher than we could accurately measure the capacitance of the membrane. Thus, we do not expect for EDL dynamics to be present in the dynamic measurements of Q-v hysteresis at frequencies below and near 1 Hz.

- In stark contrast, the system studied in the articles suggested by Reviewer 2 consists of a negatively-charged glass substrate with conical nanopores. These interfaces exhibit significantly higher values of capacitance ($\sim 10\text{nF}$), which when coupled to the higher electrolyte resistance ($\sim 0.1\text{-}1\text{G}\Omega$) within the nanopore results in significantly larger time constants of $\sim 1\text{s}$ for rearrangement of ions within the EDL. The reviewer is correct that the substrate being porous is not required for this behavior. However, the nanoscale volume of the conical pore does directly influence the solution resistance in this region, which affects the time constant for hysteretic response.

When combined with our direct evidence of dynamic changes in membrane area and thickness (and new control cases provided in Fig. S8 which show that Q-v hysteresis vanishes when EW and EC are turned-off via selection of lipids or oil), this difference in time scales for EDL dynamics provides additional proof that the dynamics and hysteresis of charge versus ac voltage in our system are controlled solely by the slow time rate changes in membrane area and thickness, and *not* due to rapid changes in EDL formed by ions on the membrane surface. We hypothesize that because of the similar time scales ($\sim 1\text{s}$) of dynamic changes in capacitance exhibited by our soft membranes, Reviewer 2 has also misinterpreted the rectification observed in our asymmetric system (caused by the difference in dipole potential that changes where EW and EC are minimized) and explained in our previous rebuttal to support the theory of EDL hysteresis.

Still, because Reviewer 2 has made interesting comparisons, we have made the following changes/additions, and included citations to the suggested articles.

Changes to the paper and SI:

- Pg 6, line 118-123 now reads:

“Subjected to a nonzero bias, $v(t)$, ions rapidly accumulate on both sides of the membrane. where they exert parallel and perpendicular forces that can affect the geometry of the interface. For a typical bilayer with capacitance of $\sim 500\text{pF}$ and aqueous droplets with solution resistance of $\sim 10\text{ k}\Omega$, the time constant for charging/discharging the electric double layer (EDL) is $\sim 5\mu\text{s}$. Therefore, at frequencies far below 20 kHz, these EDL are quasi-static with respect to ac voltage.”

- Pg 10, line 197 now includes a reference to the new data (Fig. S8) which shows no Q-v hysteresis when EW and EC mechanisms are turned off.
- Pg. 11, line 218-221 includes a new sentence with suggested references:

“Coincidentally these time constants for geometric reconfiguration are quite similar to the time constants ($\sim 1\text{ s}$) attributed to hysteretic ion rearrangement on the surfaces of charged, glass nanopores^{48,49}, despite the differences in physical origins for hysteretic charging versus ac voltage.”

- A new figure (Fig. S8) has been added to the SI to show linear, the non-hysteretic Q-v relationship that occurs for oil-free membranes that are not sensitive to EW and EC.

Reviewer #3 (Remarks to the Author):

The authors of this paper did not answer my concerns at all. Instead, they seem to have major misunderstandings about the nature of their work which makes my original assessment even more compelling.

For one, the paper by Chua the authors report in their reply does not address the issues raised by the paper in *Nanotechnology*, 24, 255201, 2013 since Chua's paper seems completely oblivious to the well-known theory of response functions. By citing Chua's paper the authors themselves seem to be totally ignorant about this fundamental theory.

Second, of course a linear capacitor in parallel to a nonlinear resistor (a leaky capacitor) would show a non-pinched hysteresis and hence divergence capacitance. That is not the point. What the *Nanotechnology* paper I was referring to discusses is that capacitances (like resistances and inductances) are derived quantities. It never discusses capacitors in parallel with resistors. I wonder if these authors have actually read it. The only measurable quantities (in the case of a capacitor) are charges and voltages. Therefore, when one extracts from these two quantities the capacitance one can get a finite, infinite, negative or positive capacitance. This is true whether one looks at it physically or just “mathematically”. There is nothing strange about this and should be well known. Sadly, neither Chua nor these authors seem to know this.

That is all that *Nanotechnology* paper is discussing. If the authors think this is “fundamentally incorrect” then they really do not understand basic concepts.

The capacitor the authors discuss in their paper is pinched because the memory arises from moving the capacitor plates and not from dielectric permittivity variations. Both are valid sources of memory.

But take for instance a capacitor whose memory comes only from a slow change in dielectric permittivity in response to the voltage (not a leaky capacitor or one with moving plates).

This capacitor will show hysteretic non-pinched q - V curves and diverging capacitance. It is a legitimate “memcapacitor” but not one showing pinched hysteresis. The memory comes from the dielectric permittivity not from a parallel resistor.

Therefore, I stand by my previous comments: there is nothing particularly novel or groundbreaking about the present work to warrant publication in *Nature Communications*.

There have been other reported cases of memcapacitors as pointed out by the authors. The authors themselves discuss the same concept in their previous publication (Ref. 16 of their paper). This new paper is yet another example of memcapacitor but not one particularly compelling.

In addition, with their reply the authors show a complete lack of understanding of basic concepts. They do not seem to realize this even after my comments.

Due to the lack of novelty and the wrong and misleading statements by the authors I therefore think this paper is not suitable for *Nature Communications*.

The mathematical definition of memory capacitance, including whether pinching in the charge-voltage plane is required, is beyond the scope of this work, and it is clear that we and Reviewer 3 disagree on this point. Interestingly, both Reviewer 3 and us agree that our assembly, which exhibits pinched hysteresis in the charge-voltage relationship, is a physical memory capacitor. Unfortunately, it appears this reviewer's personal issue with the definition has prevented a fair and thorough evaluation of our manuscript, which we believe offers multiple new contributions:

- This is the first example of a geometrically reconfigurable memcapacitor achieved via a soft, biomimetic membrane. The similarity of this structure to cell membranes found in neurons raises interesting questions regarding the possible role of variable capacitance in neural excitability.

- We demonstrated through experiments and physics-based modeling (which is rarely provided in demonstrations of memcapacitors) that this molecular assembly exhibits tunable, analog memory capacitance. While electrowetting did cause changes in membrane area in our recent publication⁴ demonstrating memory resistance with voltage-controlled ion channels, the dynamics and hysteresis in membrane capacitance coupled to both thickness and area presented herein were not quantified, modeled, or discussed in our prior work.
- Furthermore, we demonstrated herein for the first time that tuning the modular structure and composition of the membrane can enable both excitatory and depressive short-term capacitive plasticity, a result that supports integration of analog memory capacitors into adaptive circuitries inspired by the brain. Our results showing that membranes comprised of asymmetric lipid leaflets—which exhibit an intrinsic voltage difference—can exhibit depressive plasticity adds additional novelty and further highlights the modularity of our system.

Further, in our prior rebuttal/revision, we amended/replaced our original claims in the manuscript regarding significance to improve the objectivity/accuracy in our stated motivation for the work, and thus its potential significance:

“By definition, memelements are devices with resistance, capacitance, or inductance that depends on their past electrical activity⁵. While many types of memory resistors, “memristors”, have been introduced⁶⁻⁹, to date, only a sparse collection of physically realized memory capacitors¹⁰⁻²¹ and one pseudo-memcapacitor²² have been reported, despite their promise to further lower static power dissipation. Of these, most did not provide evidence of pinched hysteresis in the charge-voltage plane as proof of memory capacitance, as first defined by Chua^{5,23}, or develop realistic, physics-based models to describe and predict the state variables driving capacitive reconfigurations. Devices for which ideal, analogue memcapacitance that originates from geometrical changes in materials at the molecular scale remain unrealized to date.”

We are still confident that these statements are objective, accurate, and not over-reaching based on our careful review of the literature.

Reviewers' comments:

Reviewer #2 (Remarks to the Author):

It is a bit surprising to see the mistakes in the RC time constant calculation. For the several interfaces, which R and C should be combined if equivalent circuits are to be used, and which one/s are indeed limiting the signal and be measured?

I hope one day the authors would appreciate my previous and this final suggestion. Molecular insights could be lost when using classic engineering equivalent circuit approaches especially when the signal arises from ions/molecules. Again, would ions move across the bilayer w/ or w/o oil with a Giga Ohm seal? If not, what is the molecular origin or the charge carriers for the detected signal, particular at the bilayer region, inside & out?

Reviewer #4 (Remarks to the Author):

The authors reported a volatile, voltage-controlled memcapacitor made of biomembranes. The memcapacitance is attributed to the reversible and hysteretic geometrical changes in a lipid bilayer, with the membrane radius and thickness being the two state variables. The authors also did modeling and simulation on the non-linear dynamics of the device, and claimed that their device is truly a memcapacitor. Using biomaterials to implement memelements is of great interest in understanding the behavior of the brain which is made of organic materials. The results reported in this manuscript are hence important and the physical understanding of the system is sound. However, some minor issues should be addressed before publishing.

1. Line 118, "...ions rapidly accumulate on both sides of the membrane...". In a real biosystem, the membranes are usually leaky because of the ion channels (so that the ions will be released upon regulations from Ca^{2+} for example). Although the authors have shown in their previous work (ref. 16) that the density of ion channels are important for resistance switching, the current memcapacitor model does not consider the ion channels at all. The authors actually chose lipids with very low ion permeability in this work on purpose. It is helpful to discuss the how this model would be applicable to a real bio-membrane system. Fully integrating the ion channel is beyond the scope of this work, but a brief discussion on the 'leaky' aspect of the capacitor would be inspiring.

2. Line 122, "...these EDL are quasi-static with respect to ac voltage...". The 'quasi-static' here is with respect to ion movement, not the mechanical deformation of the lipid. It is worthwhile to clarify this in order not to generate unnecessary confusion.

3. Lines 128-131, the author discussed a case in which residual oil is trapped between bilayers, and claimed that the electro-compression does not change the bilayer area. However, the oil could be deformed and the distance between the two plates of the capacitor could be changed. This change, together with the modulation of the dielectric constant of the media resulted from the existence of the oil, could change the capacitance. Extension of the model to this case would clarify some misunderstanding in this field (deformation vs. permittivity).

4. Lines 174-175, "...the capacitance of a bilayer formed in decane is more responsive to voltage compared to one in hexadecane...". What is the fundamental reason behind this observation, a longer alkane length, a larger surface tension, a larger viscosity, or a combination of these parameters? Please explain briefly in the text.

5. In addition to the frequency dependent capacitance change as shown in Fig. 2, it would be helpful to include plots that show the evolution of other important system properties (e.g. bilayer thickness, area) as a function of the electrical stimuli in Supplementary Information.

6. Lower insets in Figs. 3a,b show the energy dissipated by the reconfigurable membrane. Please include formulas for the energy in the text or figure caption.

7. Figure 4 lower panel shows the behavior for an asymmetric bilayer, which is attributed to the polarization of molecules at the bilayer. Would this polarization change during the deformation of the bilayer?

Reviewer #2 (Remarks to the Author):

It is a bit surprising to see the mistakes in the RC time constant calculation. For the several interfaces, which R and C should be combined if equivalent circuits are to be used, and which one/s are indeed limiting the signal and be measured?

I hope one day the authors would appreciate my previous and this final suggestion. Molecular insights could be lost when using classic engineering equivalent circuit approaches especially when the signal arises from ions/molecules. Again, would ions move across the bilayer w/ or w/o oil with a Giga Ohm seal? If not, what is the molecular origin or the charge carriers for the detected signal, particular at the bilayer region, inside & out?

The reviewer continues to question if capacitive memory in our system arises from hysteretic differences in charging and discharging rates of the electric double layers that form on both sides of the membrane. While the reviewer is correct that lipid bilayer capacitance can be modeled as multiple capacitors in series, we aim to show that double layer capacitances: 1) contribute insignificantly to the total capacitance of the membrane, and 2) that the resulting charging/discharging rate (time constant) for this system is orders of magnitude faster than the rates of voltage-dependent capacitance variation that we measure.

We start by reviewing what is well known²⁻⁴ about the structure and equivalent circuit of a lipid membrane. Consider Figure R1 below, which illustrates the cross-section of a lipid bilayer without ion channels (in panel A) and the equivalent electrical circuit (in panel B). As the name suggests, the bilayer consists of two opposing lipid leaflets. However, this structure can also be viewed as a tri-layer interface, with outer polar headgroup regions (and associated electric double layers of mobile ions) sandwiching an inner hydrophobic core (~2-4 nm thick). The equivalent circuit for the system thus consists of a resistor (R_e) for the electrolyte in series with the membrane, which is modeled as a resistor (R_m) in parallel with three capacitors in series. C_{DL} represent the electric

Figure R1: A) Lipid bilayer cross-section; B) equivalent circuit that includes two capacitors for the polar regions/electric double layers and one for the inner hydrophobic core.

double layer capacitors that form as potassium (+) ions accumulate on both faces of the membrane in electrolyte media, and C_m represents the capacitance of the hydrophobic core of the bilayer.

When a voltage is applied, oppositely-charged ions are driven from the aqueous media to these opposing faces, charging the membrane. Since R_e is only c.a. 1-10k Ω , the impedance of the membrane is ~ 6 orders of magnitude higher at frequencies below c.a. 100 Hz, which causes the potential difference developed across the membrane due to ion accumulation to be nearly identical in magnitude to the applied voltage difference between electrodes. When the applied voltage is removed, excess ions retreat from the surfaces of the membrane back into the aqueous media without crossing the membrane. This is caused by the fact that R_m for a bilayer devoid of ion channels (and either with or without residual oil) is very large (10-100G Ω). Our method to induce capacitive currents using a 10Hz, 10mV triangle wave voltage also ensures that ohmic currents are not influencing our assessments of membrane capacitance. Further, we have not witnessed any electroporation events at the voltage levels that we have applied. Thus, the molecular origin of charge carriers near the membrane is the accumulation and depletion of mobile ions on each face of the capacitive bilayer, and NOT ohmic transport through the interface.

The rate of charging/discharging of the membrane is determined by the total effective capacitance of the bilayer, which is dominated by the hydrophobic core large values of double layer capacitance⁵. This reasoning stems from the fact that for an electrolyte containing 500mM KCl and a dielectric constant of 80 for the hydrated polar headgroup region, the Debye length (thickness of C_{DL}) is approximately 0.434 nm, which means C_{DL} is c.a. 138 $\mu\text{F}/\text{cm}^2$. In comparison, C_m is c.a. 0.4-0.7 $\mu\text{F}/\text{cm}^2$ (for a thickness of 2-4nm and a dielectric constant of ~ 2). For capacitors in series (as shown in Figure R1B), this difference in magnitude ($C_m \ll C_{DL}$) means that the total effective capacitance per unit area is well described by C_m . Therefore, the hydrophobic region of the membrane alone is sufficient to quantify the capacitive nature of the interface between droplets.

In terms of dynamics, this also means that the time constant for membrane charging (including the double layers) is determined by: $\tau = R_e C_m \approx 10 - 100\mu\text{s}$ (depending on the nominal area and thickness of the membrane). This is several orders of magnitude quicker than the characteristic time constants for capacitance variation, which we attribute to electrowetting and electrocompression, and is the same order of magnitude time constant we presented in the prior rebuttal. Therefore, we stand by our former estimate of double layer charging rates. Moreover, our independent evidence of changes in bilayer area between droplets that are consistent in magnitude and rates of change with those measured for capacitance provide additional proof that hysteretic double layer charging is NOT significant to the variations or memory observed.

Therefore, our story is very clear. Synthetic lipid biomembranes formed between droplets of water in oil exhibit capacitive memory caused by reversible and hysteretic changes in membrane geometry. Through experiments and modeling, we demonstrated that these geometrical changes arise from electrowetting and electrocompression which lead to an increase in membrane area and a decrease in membrane thickness, respectively as the net membrane voltage increases.

Changes to the manuscript

- Based on the reviewer's comments, we have added a sentence (lines 113-115) clarifying that the capacitance of electric double layers is not significant in our system.

- Further, we modified lines 132-135 to state that the time constants for charging both the membrane and the EDL.

Reviewer #4 (Remarks to the Author):

The authors reported a volatile, voltage-controlled memcapacitor made of biomembranes. The memcapacitance is attributed to the reversible and hysteretic geometrical changes in a lipid bilayer, with the membrane radius and thickness being the two state variables. The authors also did modeling and simulation on the non-linear dynamics of the device, and claimed that their device is truly a memcapacitor. Using biomaterials to implement memelements is of great interest in understanding the behavior of the brain which is made of organic materials. The results reported in this manuscript are hence important and the physical understanding of the system is sound. However, some minor issues should be addressed before publishing.

1. Line 118, "...ions rapidly accumulate on both sides of the membrane...". In a real biosystem, the membranes are usually leaky because of the ion channels (so that the ions will be released upon regulations from Ca²⁺ for example). Although the authors have shown in their previous work (ref. 16) that the density of ion channels are important for resistance switching, the current memcapacitor model does not consider the ion channels at all. The authors actually chose lipids with very low ion permeability in this work on purpose. It is helpful to discuss the how this model would be applicable to a real bio-membrane system. Fully integrating the ion channel is beyond the scope of this work, but a brief discussion on the 'leaky' aspect of the capacitor would be inspiring.

The reviewer poses an interesting and relevant question, since biological membranes contain channels enabling ion diffusion across the membrane. Our response first considers the effect of lower membrane resistance on total current and charge across the membrane. Via analysis of membrane impedance versus frequency, we then show how just the capacitance of the membrane can still be assessed when the resistance drops.

The current across a completely insulating lipid bilayer as described by Hodgkin-Huxley (HH) is the product of the membrane's capacitance and the rate of change in the potential with respect to time as follows:

$$I_c = C_m \frac{dV}{dt} \quad (R1)$$

The HH model, however, considers the membrane capacitance, C_m , to be constant. In our study, we show that insulating lipid membranes assembled between droplets of water in oil, exhibit voltage-dependent, hysteretic changes in capacitance caused by combinations of electrowetting and electrocompression. In this case, the total current across the membrane becomes the sum of: 1) the product of the membrane capacitance and the time rate of change in the potential; and 2) the product of voltage and the time rate change in capacitance as follows:

$$I_c = C_m \frac{dV}{dt} + V \frac{dC_m}{dt} \quad (R2)$$

Now let's **assume** that conductive ion channels (ic) are present in the membrane. The ohmic current through the channels can be calculated as follows:

$$I_{ic} = \frac{V}{R_{ic}} \quad (R3)$$

If we sum the lipid bilayer currents with the ion channel current we end up with a total current of

$$I_{tot} = C_m \frac{dV}{dt} + V \frac{dC_m}{dt} + \frac{V}{R_{ic}}. \quad (R4)$$

Equation R4 describes the resulting current across a leaky membrane (i.e., leaky capacitor). By integrating Eq. R4 with respect to time for a sinusoidal voltage bias of 150mV @ 0.17Hz, we can compute the charge across the leaky membrane, and we find that the ideal Q-V pinched hysteresis loop no longer pinches.

Figure R2A shows the computed charge stored across a purely capacitive membrane, which resembles the measurements provided in the manuscript. Figure R2B shows the elliptical Q-V, non-pinched relationship across the ion channels in the membrane assuming a membrane resistance of 100 MΩ. Figure 2C shows the combined Q-V response expected for a leaky capacitor when subjected to the same applied bias. Oppositely, increasing the membrane resistance to a higher value (100GΩ) causes the total charge stored to match that of the capacitive contribution (Figure R3).

Figure R2. Charge-voltage relationship for a membrane containing ion channels. A) The total charge across the membrane and ion channels. B) The charge stored at the capacitive membrane. C) The charge across the ion channels. The charge displayed in A is the sum of the charges displayed in B and C.

Figure R3. The charge-voltage relationship for an insulating membrane with no ion channels. Note how it is similar to the charge-voltage loop displayed in Figure R2A.

However, a leaky capacitor does not prevent the experimental measurement of the variation of the capacitance. Consider the electrical impedance of the membrane versus excitation frequency (Figure R4). This figure shows that a significant decrease in membrane resistance causes the low-frequency asymptote of the magnitude of impedance to drop and the transition frequency, f_1 , to increase for constant capacitance. The latter change means that a higher excitation frequency (for the ac triangle waveform voltage used in our study) may be required to induce capacitive current to assess memory capacitance. And importantly, adding leakage paths does not eliminate the effects of electrowetting or electrocompression on our model system. In fact, we have observed electrowetting and electrocompression affect the geometry (and capacitance) of highly-conductive membranes doped with voltage-independent gramicidin ion channels (*paper in preparation*). We

Figure R4. A Bode diagram from ref. ¹ shows the magnitude and phase of the droplet interface bilayer (DIB) as a function of frequency. A lower R_m (in the presence of ion channels) would move the transition frequency f_1 to the right.

induce capacitive currents using sufficiently high-frequency ac voltages to minimize the ohmic contribution to the measured current.

Changes to the manuscript

- We added a few sentences addressing the behavior of the system in the presence of ion channels (lines 216-218)

2. Line 122, "...these EDL are quasi-static with respect to ac voltage...". The 'quasi-static' here is with respect to ion movement, not the mechanical deformation of the lipid. It is worthwhile to clarify this in order not to generate unnecessary confusion.

Very good point raised by the reviewer. We made changes (line 134) in the text accordingly.

3. Lines 128-131, the author discussed a case in which residual oil is trapped between bilayers, and claimed that the electro-compression does not change the bilayer area. However, the oil could be deformed and the distance between the two plates of the capacitor could be changed. This change, together with the modulation of the dielectric constant of the media resulted from the existence of the oil, could change the capacitance. Extension of the model to this case would clarify some misunderstanding in this field (deformation vs. permittivity).

The reviewer raises an extremely important point and her/his suggestion is very insightful. In this manuscript we studied membranes formed using DPhPC and DOPhPC lipids. The dielectric constant of the hydrophobic chain region in such membranes is estimated to be around 2.1. In comparison, the dielectric constants of hexadecane and decane at room temperature are estimated to be around 2.09 and 1.991, respectively (Dortmund Data Bank). This means that when residual oil is present within the hydrophobic core of the membrane, the equivalent dielectric constants of the hydrophobic core is estimated to be 2.04 for DPhPC membranes in decane and 2.1 for DPhPC bilayers in hexadecane. Let's consider the decane case where electrocompression is more pronounced. Based on our experiments and simulations using the model, the maximum thinning we observed at 150 mV is around 20-25%. Since the lipid tails are incompressible, this reduction in thickness is attributed to exclusion of decane from the hydrophobic core. Even in this case, we estimate that the dielectric constant at maximum compression to be around 2.05—a change of 0.5% in capacitance which is negligible compared to the changes in capacitance caused by in thickness (3-9%) and radius (18-42%) (see Figure 5). Thus, we conclude that changes in dielectric constant for hydrophobic core of membrane are insignificant to the total response measured.

Changes to the manuscript

- We added a few sentences clarifying that the dielectric constant of the membrane changes insignificantly throughout all deformations (lines 147-151).

4. Lines 174-175, "...the capacitance of a bilayer formed in decane is more responsive to voltage compared to one in hexadecane...". What is the fundamental reason behind this observation, a longer alkane length, a larger surface tension, a larger viscosity, or a combination of these parameters? Please explain briefly in the text.

Our experiments demonstrate that a decane system responds to a wider range of frequencies compared to a hexadecane system. This means that a decane system at 1.7 Hz exhibits changes in capacitance while the capacitance of the hexadecane system at a similar frequency does not exhibit noticeable changes. The primary reason for such behavior is due to the lower viscosity of decane (~0.92 mPa.s) compared to hexadecane (~3.47 mPa.s), which enables it to respond faster. As we demonstrate in the paper, the changes in membrane capacitance are directly caused by voltage-driven increase in membrane radius and decrease in its thickness. We describe the dynamic change of both state variables with respect to time in response to ac voltage using equations 3 and 4 in the manuscript. In both equations we use effective damping and stiffness coefficients that are physically related to the dynamic viscosity of the oil. The fact that the viscosity of decane is smaller than that of hexadecane explains our findings in Table 1 which shows that for all frequencies the damping and stiffness coefficients of decane are significantly smaller than those of hexadecane. Smaller stiffness and damping coefficients automatically lead to faster response with larger oscillations to a broader range of frequencies. Also, the decane membrane at zero applied voltage is thicker, which means that larger deformations (larger changes in capacitance) are achievable.

Changes to the manuscript

- We added a clarifying sentence between lines 171-174.

5. In addition to the frequency dependent capacitance change as shown in Fig. 2, it would be helpful to include plots that show the evolution of other important system properties (e.g. bilayer thickness, area) as a function of the electrical stimuli in Supplementary Information.

Per the reviewer's recommendations we added three figures to the SI that describe the changes in membrane radius and thickness as a function of both time and voltage. Note, while we were able to measure dynamic changes in membrane radius, dynamic changes in thickness could only be simulated.

6. Lower insets in Figs. 3a,b show the energy dissipated by the reconfigurable membrane. Please include formulas for the energy in the text or figure caption.

The energy was obtained by integrating the product of the current across the membrane and the applied bias voltage over time. The current was obtained using Eq. R2.

Changes to the manuscript

- Detailed description is added to the Methods section in the manuscript (lines 415-418).

7. Figure 4 lower panel shows the behavior for an asymmetric bilayer, which is attributed to the polarization of molecules at the bilayer. Would this polarization change during the deformation of the bilayer?

The internal dipole potential in asymmetric membranes originates from anisotropic structure of the headgroup region and oriented water at the polar-nonpolar interface. Specifically, the dipole potential of a lipid bilayer consists of the following components: 1) the orientation of phospholipids carbonyls in the head group regions of the bilayer, 2) the ester linkage between the head group and the fatty acid chains, and 3) the terminal end of the fatty acid chain. As a result, dipole potential changes as lipid orientation and lateral packing change. The difference in dipole potentials between ester (DPhPC) and ether (DOPhPC) lipids that we measured by determining where capacitance is minimized (i.e. net membrane voltage is zero) was on the order of ~135-140 mV, which agrees well with the difference in monolayer dipole potentials measured independently for the same lipids via surface potential measurements⁶. Knowing that our membranes do not stretch as a result of voltage—i.e. the area per lipid is constant during electrowetting and compression-induced oil exclusion, we do not believe that the dipole potential value would change during actuation⁶.

Changes to the manuscript

None.

References

1. Sarles, S. A. Physical Encapsulation of Interface Bilayers. Dissertation, Virginia Polytechnic Institute and State University, Blacksburg, VA, 2010.
2. White, S. H.; Thompson, T. E., Capacitance, area, and thickness variations in thin lipid films. *Biochimica et Biophysica Acta (BBA) - Biomembranes* **1973**, *323* (1), 7-22.
3. White, S. H.; Chang, W., Voltage dependence of the capacitance and area of black lipid membranes. *Biophysical Journal* **1981**, *36* (2), 449-453.
4. Wobschall, D., Voltage dependence of bilayer membrane capacitance. *Journal of Colloid and Interface Science* **1972**, *40* (3), 417-423.
5. White, S. H., A Study of Lipid Bilayer Membrane Stability Using Precise Measurements of Specific Capacitance. *Biophys. J.* **1970**, *10* (12), 1127-1148.
6. Yasmann, A.; Sukharev, S., Properties of diphytanoyl phospholipids at the air–water interface. *Langmuir* **2014**, *31* (1), 350-357.

REVIEWERS' COMMENTS:

Reviewer #4 (Remarks to the Author):

The authors have satisfactorily addressed my previous concerns in the revised manuscript.